# Grey-box modeling and hypothesis testing of functional near-infrared spectroscopy-based cerebrovascular reactivity to anodal high-definition tDCS in healthy humans

**Yashika Arora[1,3], Pushpinder Walia[3], Mitsuhiro Hayashibe[4], Makii Muthalib[2,5], Shubhajit Roy Chowdhury[1]\*, Stephane Perrey[5], Anirban Dutta[3]\***

**1** School of Computing and Electrical Engineering, Indian Institute of Technology, Mandi, India, **2** Silverline Research, Brisbane, Australia, **3** Neuroengineering and Informatics for Rehabilitation Laboratory, Department of Biomedical Engineering, University at Buffalo, Buffalo, New York, United States of America, **4** Neuro-Robotics Lab, Department of Robotics, Graduate School of Engineering, Tohoku University, Sendai, Japan, **5** EuroMov Digital Health in Motion, University of Montpellier, Montpellier, France

\* src@iitmandi.ac.in (SRC); anirband@buffalo.edu (AD)

## Abstract

Transcranial direct current stimulation (tDCS) has been shown to evoke hemodynamics response; however, the mechanisms have not been investigated systematically using systems biology approaches. Our study presents a grey-box linear model that was developed from a physiologically detailed multi-compartmental neurovascular unit model consisting of the vascular smooth muscle, perivascular space, synaptic space, and astrocyte glial cell. Then, model linearization was performed on the physiologically detailed nonlinear model to find appropriate complexity (Akaike information criterion) to fit functional near-infrared spectroscopy (fNIRS) based measure of blood volume changes, called cerebrovascular reactivity (CVR), to high-definition (HD) tDCS. The grey-box linear model was applied on the fNIRS-based CVR during the first 150 seconds of anodal HD-tDCS in eleven healthy humans. The grey-box linear models for each of the four nested pathways starting from tDCS scalp current density that perturbed synaptic potassium released from active neurons for Pathway 1, astrocytic transmembrane current for Pathway 2, perivascular potassium concentration for Pathway 3, and voltage-gated ion channel current on the smooth muscle cell for Pathway 4 were fitted to the total hemoglobin concentration (tHb) changes from optodes in the vicinity of 4x1 HD-tDCS electrodes as well as on the contralateral sensorimotor cortex. We found that the tDCS perturbation Pathway 3 presented the least mean square error (MSE, median <2.5%) and the lowest Akaike information criterion (AIC, median -1.726) from the individual grey-box linear model fitting at the targeted-region. Then, minimal realization transfer function with reduced-order approximations of the grey-box model pathways was fitted to the ensemble average tHb time series. Again, Pathway 3 with nine poles and two zeros (all free parameters), provided the best Goodness of Fit of 0.0078 for Chi-Square difference test of nested pathways. Therefore, our study provided a systems biology approach to investigate the initial transient hemodynamic response to tDCS based on fNIRS tHb data. Future studies need to investigate the steady-state responses, including steady-

**Funding:** The computational research was funded by the Science and Engineering Research Board, a statutory body of the Department of Science and Technology, Government of India, and Ministry of Electronics and Information Technology, Government of India (YA); Indian Institute of Technology Mandi, India (SRC); Community for Global Health Equity at the University at Buffalo, USA (AD). The human subject research was funded by the LabEx NUMEV, France (MM, MH, SP, AD). The funders had no role in study design, data collection and analysis, decision to publish, or preparation of the manuscript.

**Competing interests:** The authors have declared that no competing interests exist.

state oscillations found to be driven by calcium dynamics, where transcranial alternating current stimulation may provide frequency-dependent physiological entrainment for system identification. We postulate that such a mechanistic understanding from system identification of the hemodynamics response to transcranial electrical stimulation can facilitate adequate delivery of the current density to the neurovascular tissue under simultaneous portable imaging in various cerebrovascular diseases.

## Author summary

Non-invasive brain stimulation techniques, including transcranial direct current stimulation (tDCS), is increasingly being used for the neuromodulation of human brain; however, the vascular mechanisms of action have not been investigated systematically. We applied a rational computational modeling approach to human portable neuroimaging data and found that the tDCS effects in the tissues surrounding a blood vessel explained the blood volume changes, called cerebrovascular reactivity to tDCS, during the first 150 seconds of anodal HD-tDCS in eleven healthy humans. In this study, we limited our investigation to initial hemodynamics response to non-invasive brain stimulation. Future studies need to investigate the non-invasive brain stimulation effects on steady-state oscillations where transcranial alternating current stimulation may be effective in entraining beneficial oscillations. Such a mechanistic understanding from system identification of the hemodynamic responses based on portable neuroimaging can facilitate adequate delivery of the current density to the neurovascular tissue where hypoperfusion has been associated with cerebrovascular diseases, including cognitive impairment. Furthermore, low-cost wearable portable neuroimaging approach in conjunction with non-invasive brain stimulation is amenable to point of care settings including home-based therapy.

## Introduction

Cerebral blood flow (CBF) regulation is crucial for normal brain activity where hypoperfusion has been associated with cerebrovascular diseases, including cognitive impairment in various cross-sectional studies [1]. Cerebrovascular disease refers to conditions that have an effect on blood vessels and blood supply to the brain [2]. Since vascular factors are an important contributor to cerebrovascular disease, including a role in mild cognitive impairment and dementia [3], that is predicted to increase to 152 million by 2050 [4]; therefore, therapeutic measures for the cerebrovascular disease are crucial. Transcranial electrical stimulation (tES), particularly transcranial direct current (tDCS), has been shown to be a promising therapeutic method that can evoke regional CBF [5], which may be able to ameliorate hypoperfusion and the related cognitive impairments. Here, CBF is regulated primarily by three mechanisms, cerebral autoregulation that maintains the CBF under changes in systemic blood pressure; cerebral vasoreactivity that is the response to the arterial $CO_2$ partial pressure changes; and neurovascular coupling that is the response to the neuronal activity [6]. However, the physiological mechanisms of tDCS evoked CBF response [5,7–9] are unknown. A recent study [9] showed that the spatial distribution of CBF changes correlated with the tDCS-induced electric field distribution ($< 1$ V/m) computed using finite element modeling. CBF changes can also be evoked rapidly ($<100$ ms) with transcranial alternating current stimulation (tACS) at 10–20 Hz; however, at higher electric field strengths (5–20 V/m)[10]. Since neurovascular coupling related

hemodynamic response should start about 600 ms following the stimulus based on an experimental study by Devor et al. [11]; therefore, such rapid changes in the CBF due to tACS is postulated to indicate the direct effect of the electric field on the vascular neural network [7,8,12] (e.g., perivascular nerves, neuronal nitric oxide expressing interneuron [13]). The proximal pial arteries and the descending arteries have the fastest onset time followed by the capillaries (spatiotemporal characteristics of pial, penetrating, and micro-vessels are summarized in Schmid et al. [14]), where direct neurocapillary modulation by tES may also be possible [15]. Consequently, the resultant spatiotemporal dynamics of the vascular response to electrical stimulation can be quite complex due to the interdependence of the nested spatiotemporal dynamics of the pial arteries, descending arteries, and the capillaries. Therefore, individual hemodynamic effects of the tES current density via various neurovascular pathways need to be investigated using mechanistic model-based hypothesis testing where CBF responses can be site-specific and subject-specific [16].

In this study, we investigated the physiological mechanisms of tDCS evoked CBF response based on grey-box modeling and hypothesis testing. Experimental studies have indicated that long-term tDCS tends to transform neuronal activity through an induced electric field modulation of the cortical neurotransmitters (like gamma-aminobutyric acid and glutamate) during tDCS [17,18]. Besides, there is evidence of studies on CBF modulation through tDCS [5]. The neuromodulatory consequences of tDCS are understood to be generated due to the induced electric field (and current density) in the cortex by applying a weak direct current through scalp electrodes, causing cortical excitability changes [19]. It is evident from neurophysiological studies that the induced electric field can change neuronal excitability with current intensities ranging from 0.7 to 2.0 mA over 9–20 minute sessions [20]. The current applied by surface electrodes in tDCS is shunted through the scalp and cerebrospinal fluid (CSF) and only a fraction of the current reaches the cortex, producing a weak electric field (~0.3 V/m per 1mA of applied current) [21,22] that can subthreshold polarize the neurons. However, persistent (>9 min) weak electric field can lead to neuroplastic changes and excitability after-effects, postulated to be driven by persistent calcium flux, which in turn can affect cortical excitability [23], alter the firing rate of neurons [24,25], and modify spatiotemporal brain networks related to information transfer in the brain [26]. So, a majority of research on tDCS has focused on cortical neuronal after-effects following long duration (>9 min) weak electric field stimulation [17]; however, investigation of the immediate effects of the electric field on all neurovascular targets in the cortical tissue may accord to better understanding of the vascular neural network [12] mechanisms to hemodynamic response. For example, Guhathakurta and Dutta [27] postulated based on finite element modeling of the electric field strength that the pial arteries (and arterioles) contain perivascular nerves within their adventitial layer that can be strongly affected by tDCS-induced electric field. This was due to the magnitude and spread of the tDCS current density in the CSF that surrounds pial vessels which can be much higher than that in the brain parenchyma–details in the S2 Text and S4 Fig. Then, autonomic nerve fibers, including noradrenergic perivascular axons, richly innervate the cerebral vessels within their adventitial layer, especially the larger arteries including pial vessels, and tDCS electric field can affect the limits of vasomotor control [28] and the autoregulation plateau [29]. Autoregulation is important since pial arteries start the pressure-driven blood pathway to the cortex and have a robust network topology that guarantees a constant blood supply (reviewed in Schmid et al. [14]). Also, the pial arterial network structure on the cortical surface is comparable to that of a honeycomb [30] that can cause distortion of the current flow from the CSF into the gray matter (GM) due to the large differences in the conductivity [31] of the CSF, blood vessels, and GM. Such current flow distortion from the CSF into GM, and then around the penetrating

blood vessels [15], may be responsible for affecting the vascular neural network [12] while also distorting the electric field in the gray matter.

In this study, we considered a mechanistic model-based hypothesis testing of the coupling mechanisms where neuronal and vascular functions are closely interconnected through neuro-vascular mechanisms, as evident from studies using multimodal imaging techniques [32–34] like functional magnetic resonance imaging (fMRI), electroencephalogram (EEG), magnetoencephalogram (MEG) and, more recently, functional near-infrared spectroscopy (fNIRS) [35]. fNIRS provides a portable optical imaging technology that can be conducted in conjunction with transcranial electrical stimulation without interference. Therefore, combining tDCS with fNIRS is feasible to capture variable neurovascular and neurometabolic effects; however, physiologically guided mechanistic models are necessary for hypothesis testing in systems biology. Such mechanistic understanding is crucial for clinical applications where our prior case series in chronic (>6 months post-stroke) ischemic stroke identified a prolonged initial dip (12.13±3.8sec) in oxyhemoglobin concentration as a response to anodal tDCS in one subject who also complained of headache with throbbing pain following 15 min of anodal tDCS [36]. In fact, headache after tDCS and repetitive transcranial magnetic stimulation (rTMS) is quite common (11.8% in tDCS and 23% in rTMS) adverse event that needs further investigation [37] from neurovascular perspective. An integrated system of neurons, astrocytes, and vascular cells form a neurovascular unit (NVU) that facilitates brain homeostasis and links neuronal, metabolic, and vascular activities at cellular level [38]. So, tDCS can directly affect the excitability of the cortical neurons and can also have an indirect effect via vascular neural network [12] and neuronal nitric oxide (nNOS) expressing interneuron [13] or modulatory effects on non-neuronal cells [39] due to perivascular microstructure [15].

Mathematical models of NVU are well published in fMRI, which can span from completely phenomenological to detailed mechanistic models (described in the Methods section). Data-driven black-box systems approaches provide a correlate of neural and hemodynamic response at an abstract level under the assumption of neurovascular coupling at the cellular level; however, such black-box systems approaches do not aim to explicitly capture the underlying cellular mechanisms of action. We have presented a phenomenological model for capturing cerebrovascular reactivity to anodal tDCS based on solely neuronal effects [40,41]. In this study, tDCS-induced current density in the neurovascular tissue is postulated to affect not only the neuronal cells but also the astrocyte and the cerebral blood vessels, composed of pericytes, smooth muscle cells and endothelial cells. Therefore, we aimed for model fitting and model evidence of the modulatory consequences of tDCS on blood vessels that can be via neuronal and non-neuronal cells. A deeper understanding of the signaling pathways inside the NVU is essential for a mechanistic understanding of tDCS-induced electric field effects on the hemodynamics in health, aging, and disease [28] that we called cerebrovascular reactivity (CVR) to tDCS [40].

In our grey-box model, CVR is based on a change in the blood circumference (and blood volume) in response to tDCS. In healthy tissue, CVR is a compensatory mechanism where blood vessels dilate in immediate response to the vasodilatory stimulus to regulate the resistance to the flow, modifying the cerebral perfusion [42]. We postulated that the immediate hemodynamic response [10] based on CVR to short-duration tDCS can provide a marker of blood vessels' capacity to dilate that can be hampered in various cerebrovascular diseases [43,44]. However, longer duration (>9 min) tDCS will induce neuroplastic changes [17] where non-linear bidirectional neurovascular interactions [45] can make the mechanistic models too complex for fNIRS data fitting and hypothesis testing. Therefore, the current study investigated the initial CVR to tDCS by grey-box linear modeling of the intimate relationship between neuronal activity and hemodynamic response that involved the signaling pathways

across the lumped model of the NVU compartments. Although finite element modeling has indicated neurocapillary modulation by tDCS [15] through the interaction of the electric field with the NVU; however, detailed neurocapillary finite element modeling becomes challenging with uncertainties in the model parameters due to the folded cortical structures and complex vascular network. Therefore, finite element modeling [46] without model tuning [47] to fit individual response, captured with functional imaging data, can produce unpredictable and variable tDCS effects across subjects that is already evident [48,49].

In this mathematical modeling and hypothesis testing paper, our systems biology investigation is based on the dynamics within the four compartments of the NVU: synaptic space, intracellular astrocyte space, perivascular space, and intracellular space of the arteriolar smooth muscle cell (SMC). We analyzed the pilot data from healthy subjects from our prior work [50] to evaluate the role of various signaling pathways within NVU using a grey-box linear systems identification approach. In a classical neurovascular coupling response, the initial phase includes an increase in deoxy-hemoglobin concentration (deoxy-Hb) and a decrease in oxy-hemoglobin concentration (oxy-Hb) that precede an overall rise in blood volume [11]. Here, low-oxygen feedback regulation in "metabolic hypothesis" states that an increase in neuronal synaptic activity causes additional energy and oxygen demand (i.e., an increase in deoxy-Hb and decrease in oxy-Hb), causing various vasodilation agents to send signals to cerebral vasculature for vasodilation, resulting in an increase in regional CBF and oxy-Hb concentration. This metabolic hypothesis can also explain the initial dip in oxy-Hb concentration [51] in the period immediately following tDCS before reaching peak levels, as observed in our previous study on ischemic brain [36]. In healthy subjects, Muthalib et al. [52] found the group averaged initial dip in oxy-Hb within the first 15s of high-definition tDCS. An increase in oxy-Hb immediately following tDCS may be explained through a feedforward "neurogenic hypothesis," whereby the direct neuronal modulation by tDCS causes a discharge of various vasoactive agents and an increase in oxy-Hb [53]. Here, autonomic and sensory nerves from cranial ganglia and intrinsic innervation of the cerebral microcirculation [54] can be affected by tDCS. We now know that the nerve fibers from the ganglia belonging to the sympathetic, parasympathetic, and sensory nervous systems innervate the intracranial blood vessels [54]. Then, intracerebral blood vessels [54] are surrounded by astrocytes [55] that can communicate tDCS effects [56] on the pyramidal neurons [41] ("neurogenic hubs" of the NVU) to the blood vessels. The "neurogenic hypothesis" can also be applied to the transmural electrical stimulation of the perivascular nerves [28] (e.g., neuropeptide Y is an important vasoconstrictor [57] of sympathetic innervation, parasympathetic innervation for vasodilation), and nNOS expressing interneuron (e.g. NO for vasodilation [58]) in the intracranial blood vessels [54] viz. pial arteries and arterioles [27] that can have a complex compounding effect to the blood volume (deoxy-Hb+oxy-Hb) and CVR including initial dip in the blood volume due to vasoconstriction [57]. An initial dip in the blood volume captured with fNIRS total haemoglobin concentration (tHb) based CVR to tDCS was found in few healthy subjects in prior work [52]. Here, tDCS effect on the blood vessels is postulated to be via transmural stimulation of perivascular neurotransmitters that can also promote noradrenaline release. Also, tDCS may affect the sensory fibers involved in cranial pain syndromes that stores calcitonin gene-related peptide (CGRP), substance P, neurokinin A, NOS, and nociceptin, inter alia [54]. Somatic afferents [59,60] including extrinsic perivascular innervation [61] may also be responsible since tingling sensation (70.6%) and light itching sensation under the stimulus electrode (30.4%) are quite common adverse events that were observed even with short-duration sham tDCS [37], viz. trigeminal nerve stimulation with supraorbital tDCS electrode is possible due to electrode edge-effects [62]. Here, we postulate that this complex neurovascular response can be elucidated with fNIRS combined with EEG at a high temporal resolution [50].

Of the available functional neuroimaging technologies, fNIRS, a portable, noninvasive clinically available tool, allows monitoring the local cortical hemodynamic response at the point of care with reasonable spatial resolution and better temporal resolution than the gold standard of fMRI. Even low-density fNIRS can provide a measurement of changes in the cerebral oxygenation (oxy-Hb and deoxy-Hb) and blood volume (sum of oxy-Hb and deoxy-Hb) [63,64] that is a promising tool to evaluate CVR to tDCS as evident from related studies [65,66]. The modality has been extensively used in various brain diseases like epilepsy, stroke, Parkinson's disease, and mild cognitive impairment [67,68]. It can also provide an indirect measure of CBF during tDCS where the advantage over other neuroimaging modalities like fMRI and PET are: portability, better safety, higher temporal resolution, and cost effectiveness [69]. Also, fNIRS can be combined with tDCS with no electro-optic interference to measure hemodynamic response during electrical stimulation. Here, dosing of the tDCS-induced current density can be monitored for safety in a diseased state since the blood-brain barrier (BBB) dysfunction [70] can be worsened by an increased BBB permeability [71]. In principal accordance, the current study considered low-density fNIRS based measure of tHb changes (related to blood volume changes [72]), obtained by adding oxy-Hb and deoxy-Hb changes (tHb = deoxy-Hb+oxy-Hb), during tDCS to account for the vessel volume changes to fit in our mathematical models for hypothesis testing. Here, tHb change was considered as a better measure of the change in the regional cerebral blood volume [72] compared to oxy-Hb or deoxy-Hb content taken individually. The signaling pathways modulating vascular response to the electric field in the perivascular space, astrocytes, and vascular SMCs were also evaluated in addition to the neuronal pathway [40] and compared using grey-box linear systems' transfer function analysis in eleven healthy subjects from our prior experimental studies [50,52]. In our previous works, Muthalib et al. [52] found a greater increase in oxy-Hb "within" than "outside" the spatial extent of the $4 \times 1$ high-definition (HD)-tDCS electrode. In the current study, we used that experimental data in conjunction with our novel grey-box linear model to investigate tHb as the hemodynamic correlate of the current density effects "within" the spatial extent (called the "targeted-region") as well as "outside" the spatial extent (called the "nontargeted-region") of the $4 \times 1$ HD-tDCS electrode. At both the targeted-region and the nontargeted-region, the increase in the blood volume during HD-tDCS was considered to be caused by a mix of metabolic and neurogenic factors that were mathematically formulated for grey-box linear systems identification. Here, grey-box linear systems identification was performed following model linearization of a physiologically detailed NVU model to find appropriate model fit (mean square error) and model complexity (Akaike information criterion [73]) to the tDCS-evoked change in the fNIRS-tHb data.

## Methods

### Ethics statement

Institutional Review Board of EuroMov (University of Montpellier, France) provided approval for all procedures performed involving human participants. Informed written consent was obtained from the human subjects voluntarily participating in this study in accordance with the Declaration of Helsinki.

### (A) Subjects and the experimental protocol

Informed written consent was obtained from eleven healthy human subjects (1 female, 19–45 years old) who voluntarily participated in this study in accordance with the Declaration of Helsinki. The subjects had no known neurological or psychiatric history, nor any contraindications to tDCS. In this study, the pilot data from our prior work on online parameter

estimation with an autoregressive model [50] was analyzed with a grey-box model for a mechanistic understanding of the tDCS action on the total hemoglobin concentration.

During the experiment, the subjects were comfortably seated with eyes-open in an arm chair with adjustable height. The set-up of the high-definition (HD) tDCS electrodes and fNIRS optodes were mounted on the surface of the scalp according to the 10/10 system (see Fig 1). The anodal HD-tDCS (StarStim, Neuroelectrics NE, Barcelona, Spain) was configured in a 4 × 1 ring montage with the anode placed in the center (C3) in a region overlying the left

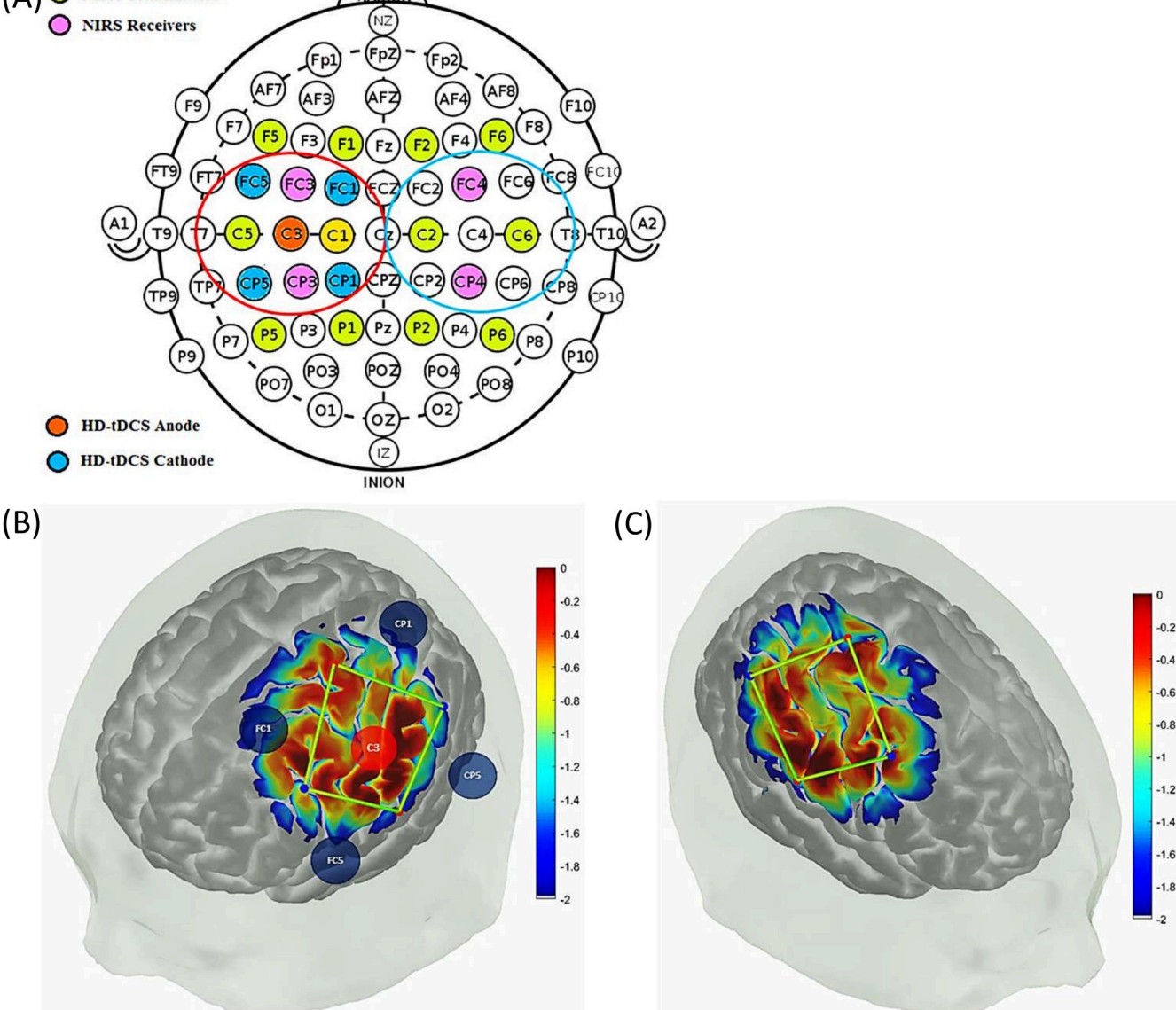

**Fig 1.** (A) HD-tDCS-fNIRS montage on a 10/10 EEG system. The fNIRS time series data from the transmitter-receiver optode combinations in the vicinity of 4x1 HD-tDCS electrodes (fNIRS channels encircled in red, left hemisphere: C5-CP3, C1-CP3, C5-FC3, and C1-FC3), i.e. the targeted-region, as well as the non-targeted region in the contralateral hemisphere (fNIRS channels encircled in blue, right hemisphere: C2-FC4, C2-CP4, C6-FC4, and C6-CP4) were used for the grey-box modeling. (B) Measurement sensitivity of the fNIRS channels at the grey matter on the left hemisphere along with the HD-tDCS electrodes in the Atlas Viewer open-source package (https://github.com/BUNPC/AtlasViewer). The sensitivity values are displayed logarithmically by the color scale. (C) Measurement sensitivity of the fNIRS channels at the grey matter on the right hemisphere in the Atlas Viewer open source package (https://github.com/BUNPC/AtlasViewer). The sensitivity values are displayed logarithmically by the color scale.

primary sensorimotor cortex. The return electrodes were placed approximately 4 cm away at FC1, FC5, CP5, and CP1 in the 10/10 system (see Fig 1). In the rest of the document, 'HD-tDCS' means 'anodal HD-tDCS' since we only performed anodal tDCS for this human pilot study. Measurements of changes in eyes-open resting-state oxy-hemoglobin (oxy-Hb) and deoxy-hemoglobin (deoxy-Hb) concentrations from the bilateral primary sensorimotor cortex were made from a 16-channel continuous-wave NIRS system (OxymonMkIII, Artinis Medical Systems, Zetten, The Netherlands) at a sampling frequency of 10 Hz. The receiver-transmitter distance of 3 cm was based on our prior work [50]. The receivers (Rx) were placed on the FC3and CP3 for the left hemisphere and FC4 and CP4 for the right hemisphere. Transmitters (Tx) were placed diagonally, i.e., at P1, P5, C1, C5, F5, and F1 for the left hemisphere, and P6, P2, C6, C2, F2, and F6 for the right hemisphere, as shown in Fig 1. The HD-tDCS was conducted using Pistim (Neuroelectrics NE, Barcelona, Spain) electrodes (contact area 3.14cm$^2$) over the left primary motor cortex region to deliver 2mA current for 10 minutes, with a ramp up and ramp down of 30 seconds. Eyes-open resting-state data for changes in oxy-Hb and deoxy-Hb was recorded. The effect of HD-tDCS on blood volume was considered for the grey-box linear modeling, so the average values of tHb (= oxy-Hb+deoxy-Hb) were obtained from the optodes in the vicinity of 4x1 HD-tDCS electrodes: C5-CP3, C1-CP3, C5-FC3, and C1-FC3, i.e., the targeted-region as well as the nontargeted-region from contralateral hemisphere (C2-FC4, C2-CP4, C6-FC4, and C6-CP4). The first criterion to confirm the signal quality was a visual inspection for the presence of cardiac pulsation, in either the time or the frequency domain (peak around 1 Hz cardiac frequency).

Due to the lack of short-separation channels to perform short source-detector regression to remove extra-cerebral hemodynamics, we performed data-driven principal component analysis (PCA) to identify the signal components that explained the greatest amount of covariance across all the spatially symmetrically distributed 16 channels. This pre-processing of the fNIRS data was performed with HOMER2 (v2.2) routines (hmrIntensity2OD and hmrOD2Conc, respectively) using the modified Beer-Lambert law and their standard pipeline. Specifically, motion correction (hmrMotionCorrectWavelet) and zero-phase bandpass filtering ("hmrBandpassFilt") was performed to extract the frequencies between 0.01 Hz and 0.1 Hz across all the 16 channels. Here, the PCA approach was applied to improve the signal to noise ratio towards neurovascular and neurometabolic coupling than systemic physiology [74]. The neurovascular and neurometabolic coupling-related hemodynamic response should lead to an initial increase in deoxy-Hb and an equal decrease in oxy-Hb. Then, the blood volume (measured with tHb) should start to increase about 600ms following the neural stimulus [53]. Such differential activation of oxy-Hb and deoxy-Hb over longer timescale (600 sec; see S2 Table) was found in the vicinity of 4x1 HD-tDCS electrodes: C5-CP3, C1-CP3, C5-FC3, and C1-FC3, i.e., the targeted-region, that indicated neurovascular coupling related hemodynamic response by HD-tDCS current density at the ipsilateral primary motor cortex. Therefore, the fNIRS channels were analyzed for the negative correlation between oxy-Hb and deoxy-Hb dynamics over 600 sec for each subject that is based on the rationale for correlation-based signal improvement [75] for the contrast to noise ratio (see S2 Table). We also investigated the fNIRS time series data at the nontargeted-region in the contralateral sensorimotor cortex where the grey matter was not directly affected by the HD-tDCS current density–confirmed using finite element modeling (see Fig 2A). Current spread (see S4 Fig) in the highly conductive CSF (mean 1.69 S/m [31]) to tissue boundary compared to grey matter (mean 0.60 S/m [31]), blood (mean 0.58 S/m [31]), and vessel wall (0.46 S/m [31]) can lead to a spatial change in the tangential surface current ($J_{Tin}$), e.g., along the pial surface (see Fig 2C), resulting in the activating function for the perivascular nerves. We have postulated [50] the propensity of HD-tDCS current density to affect the autonomic nerves in the adventitial layer–see Fig 2B. Then, complex

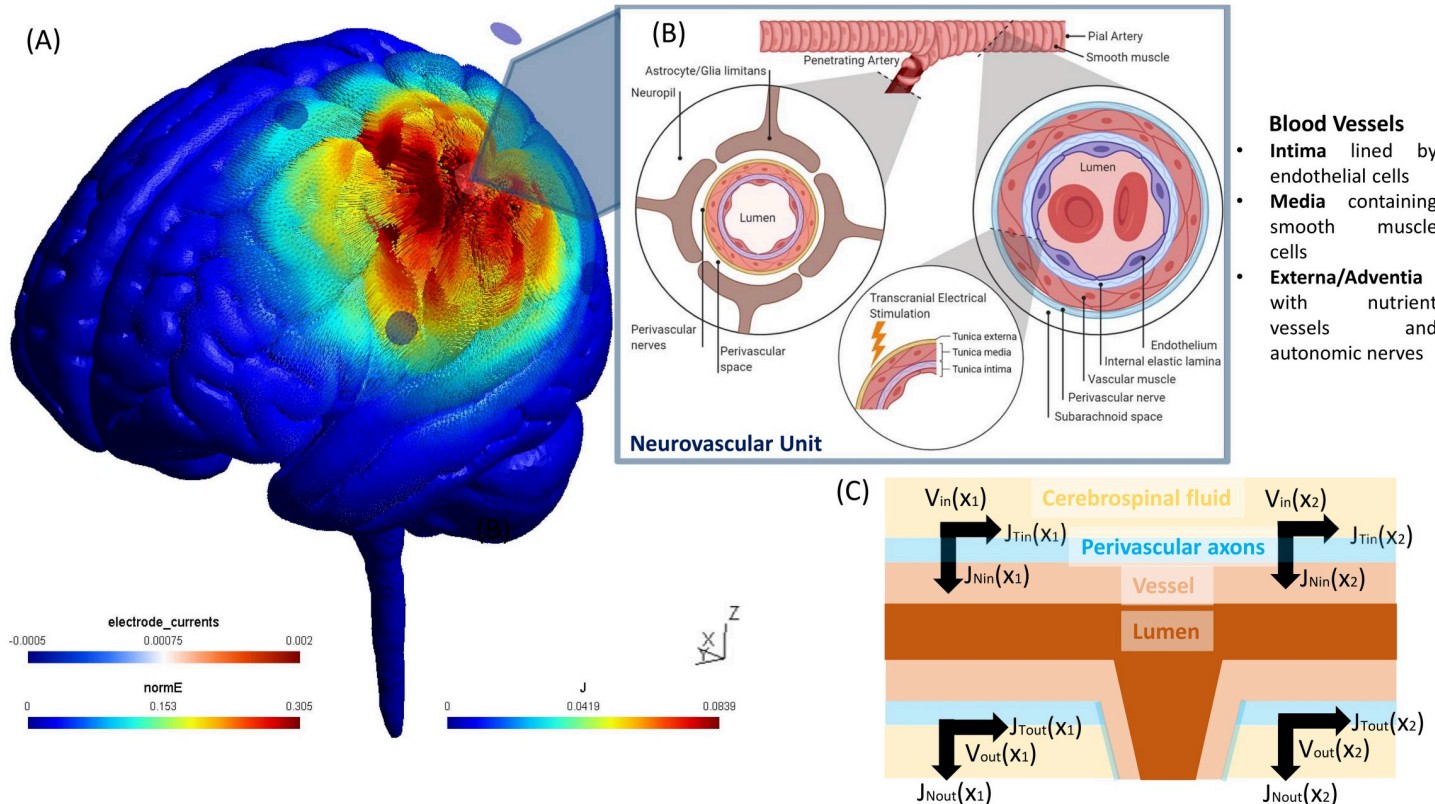

**Fig 2. The illustration shows transcranial electrical stimulation induced electric field (E) and its current density (J) in the highly conductive cerebrospinal fluid (CSF) around blood vessels penetrating from the subarachnoid space into the Virchow Robin spaces or the perivascular spaces.** Then, the extracellular electric potential (V), a function of electric field (E), can influence the perivascular nerves in the perivascular spaces thereby affecting the neurovascular unit illustrated with BioRender software (https://biorender.com/). Here, the pial arterioles have thick layer of smooth muscle cells that are surrounded by CSF in the subarachnoid space. So, the extracellular field can affect the smooth muscle cells of the vessel via its densely innervated nerve fibers originating from cranial autonomic and sensory ganglia, such as the sympathetic, parasympathetic and trigeminal ganglia. (A) Electric field strength (in V/m) along with the current density vector (in A/m$^2$) computed at the gray matter surface due to 2 mA anodal HD-tDCS (4 x 1 configuration) over left sensorimotor cortex using finite element modeling in the SimNIBS open source package (https://simnibs.github.io/simnibs/build/html/index.html). (B) Blood vessels are composed of endothelial cells (in the intima layer), smooth muscle cells (in the medial layer), and extracellular matrix (containing the perivascular nerves), where the electric field can affect the smooth muscle cells and the perivascular nerves. (C) Perivascular nerves on the surface of pial vessels, on the surface of the brain are particularly susceptible to tangential surface current (J$_{Tin}$) due to the current spread in the highly conductive (mean 1.69 S/m) cerebrospinal fluid (CSF) boundary compared to grey matter (mean 0.60 S/m), blood (mean 0.58 S/m), and vessel wall (0.46 S/m). Spatial change in the J$_{Tin}$, e.g., along pial artery (X$_1$, X$_2$), can lead to the activating function that is proportional to the second spatial derivative of the extracellular potentials (V$_{in}$, V$_{out}$) along the axon.

bidirectional communication between astrocytes and vascular reactivity governs the arteriolar calcium oscillations [76] that can be captured with a detailed physiological NVU model (see Section C). In this study, we limited our investigation to the initial transient response within the neurovascular coupling frequency band (0.01–0.05Hz) of the fNIRS data for NVU model fitting within the first 150sec of the HD-tDCS (30sec ramp-up and 2min steady-state). Here, any low-frequency (0.01–0.05Hz) steady-state vessel oscillations were considered related to the interplay between the neurovascular coupling and the vasomotor control [77–79].

## (B) fNIRS-tHb response during HD-tDCS (experimental data):

To determine fNIRS-tHb response, the mean value from the baseline (before HD-tDCS) was subtracted and then the time course was normalized with the maximum (the maximum value gets transformed into a 1) for the changes in total hemoglobin concentration changes (fNIRS-tHb response). Fig 3 shows the change in the normalized tHb from baseline (mean of 2 min

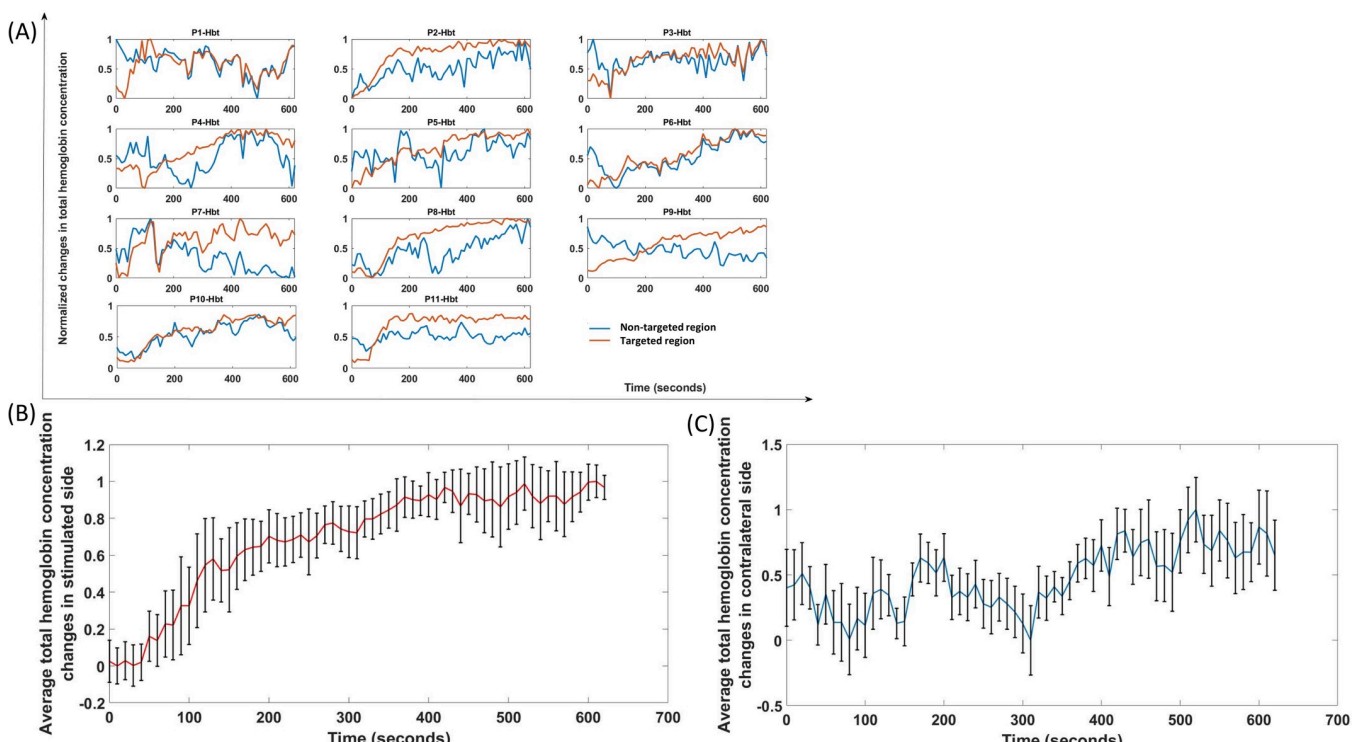

**Fig 3.** (A) Normalized time course of changes in the total hemoglobin concentration during anodal HD-tDCS (2mA current for 600 seconds, with ramp-up period of 30 seconds) at the targeted-region (red) and the nontargeted-region (blue) of eleven healthy subjects (P1-Hbt to P11-Hbt). (B) Ensemble averaged HD-tDCS evoked changes in total hemoglobin concentration (mean±standard deviation) at the targeted-region. (C) Ensemble averaged HD-tDCS evoked changes in total hemoglobin concentration (mean±standard deviation) at the nontargeted-region.

before tDCS perturbation) obtained during anodal HD-tDCS stimulation from fNIRS channels around the 4x1 HD-tDCS anode, i.e., the targeted-region and from the nontargeted-region at the contralateral sensorimotor cortex (shown by the montage in Fig 1). The 0-150sec time course of changes in tHb at the targeted-region showed a steeper increase compared to the nontargeted-region that is more evident in the ensemble averaged time courses of tHb (Fig 3B and 3C) at the targeted-region and at the nontargeted-region respectively across subjects.

## (C) Detailed physiological model review and selection

Various mathematical models representing the NVU mechanisms with major compartments–neuron, astrocyte, and vascular cells–have been published. A description of these published mathematical models is given in Table 1. These models are constructed using physical, electrical, and hemodynamics laws tied with experimental analysis to demonstrate the signaling in different NVU cell constituents. The inter-compartmental dynamics are represented by a set of differential equations having compartment-specific variables. The first models were presented by Riera et al., 2005 [80,81] for fusing EEG and fMRI data as listed in the Table 1. Then, various models were presented by Bennett et al. [82] and Farr and David [83] that used synaptic space, astrocyte, smooth muscle cell, perivascular space and endothelial cells as components of a neurovascular system. Their studies presented the coupling between neuronal activation (through K+ and glutamate) and arteriolar dilation via astrocytic potassium ions (K+), epoxyeicosatrienoic acids (EETs), and calcium ion (Ca2+) channels. Then, studies by different

**Table 1. Summary of studies detailing mathematical models of the neurovascular unit (NVU) components and their input (I) and output (O).**

| S. No. | Study | NVU system | Description |
|---|---|---|---|
| 1 | Riera et al., 2005 [80] | Synaptic space, neuron-astrocyte interface, capillary, post capillary; I: stimulus (afferent pathways); O: vessel volume | The study used a bottom-up physiological model for fusing EEG and fMRI data depicting the electro-vascular coupling. Synaptic activity and hemodynamics were governed by mesoscopic dynamic equations system. |
| 2 | Riera et al., 2006 [81] | Canonical neural mass component, electrovascular coupling component, vascular state dynamics component; I: excitatory synaptic inputs; O: Cerebral blood volume and cerebral blood flow | The study presented the cerebral architecture, electro-vascular coupling and energy considerations with respect to EEG and fMRI data fusion. The study extended balloon approach for modeling the vascular changes. |
| 3 | Bennett et al., 2008 [87] | Synaptic space, astrocyte, SMC; I: Glutamate released from glutamatergic synapse; O: Cerebral blood volume and cerebral blood flow | The study showed that the coupling between glutamatergic synapses to arteriolar SMC is mediated by astrocytic epoxyeicosatrienoic acids (EETs). Results depicted a linear rise in blood flow with an increase in numbers of activated astrocytes; however, the response was non-linear with respect to the release of glutamate. |
| 4 | Banaji et al., 2008 [88] | Cerebral circulation component, mitochondrial metabolism component; I: Blood pressure changes, changes in arterial oxygen and carbon-dioxide levels, functional activation; O: cerebral metabolic rate of oxygen (CMRO2), changes in oxidation level | The study modeled the brain circulation and metabolism to analyze the experimental fNIRS data in response to various stimuli. |
| 5 | Farr and David, 2011 [83] | Synaptic space, astrocyte, perivascular space, SMC, and endothelial cells; I: Glutamate and $K^+$ in synaptic space; O: arteriolar diameter | The study showed that the coupling between neuronal activation (due to $K^+$ and glutamate) to arteriolar dilation was mediated by astrocytic $K^+$, EET, and $Ca^{2+}$. |
| 6 | Witthoft and Karniadakis, 2012 [45] | Synaptic space, astrocyte, perivascular space, SMC; I: Glutamate and $K^+$ in synaptic space; O: arteriolar radius | The study showed the bidirectional communication between cerebral astrocytes and the microvessels. Major signaling pathways considered were: neural synaptic $K^+$ and glutamate to astrocytes, $K^+$ signaling between astrocytes and microvasculature, and microvasculature to astrocytes via astrocyte perivascular endfoot. |
| 7 | Chander and Chakravarthy, 2012 [89] | Neuron, astrocyte, vessel, and interstitium; I: Synaptic current and Adenosine Triphosphate (ATP); O: vessel radius via EET signaling | The study presented a model for the neuron-astrocyte-vessel loop based on neuronal and metabolic activity. |
| 8 | Witthoft et al., 2013 [90] | Synaptic space, astrocyte, perivascular space, SMC; I: Glutamate and $K^+$ in synaptic space; O: arteriole radius | Extended version of the model from Witthoft and Karniadakis, 2012[42] with $K^+$ buffering across all components of NVU. |
| 9 | Chang et al., 2013 [91] | Soma, dendrite, extracellular space, vascular tree compartment, glial compartment; I: extracellular $K^+$; O: vessel radius | The study demonstrated the coupling between the vascular diameters and neuronal activity mediated by $K^+$ concentrations in extracellular space in the vicinity to dendritic processes that were assumed to be buffered through astrocytes. |
| 10 | Dormanns et al., 2015 [92] | Neuron, synaptic cleft, astrocyte, perivascular space, SMC, endothelial cell, and arteriolar lumen; I: synaptic $K^+$; O: arteriole radius | The study used lumped model of the NVU for depicting the connection between a neuron and the perfusing arteriole through the astrocytic perivascular $K^+$ signaling and the SMC's $Ca^{2+}$ dynamics. The study showed the significance of luminal agonists in flowing blood influencing the endothelial and SMC dynamics. |
| 11 | Dormanns et al., 2016 [84] | Neuron, synaptic cleft, astrocyte, perivascular space, SMC, endothelial cell, and arteriolar lumen; I: synaptic $K^+$; O: arteriole radius | An extended version of Dormanns et al., 2015 model[67] with NO signaling pathway. The model considered the production of NO in the neuron and the endothelial cell compartments and its diffusion in the other compartments. |
| 12 | Blanchard et al., 2016 [93] | Pyramidal cells, interneurons, extracellular space, astrocytes, vessels; I: Excitatory postsynaptic potential (EPSP) and Firing rates (FR) from pyramidal cells and interneurons; O: local field potential and regional cerebral blood flow | The study demonstrated the connection between the neuronal activity and regional CBF via neuro-glio-vascular link at the population scale (voxel). The model evaluated the role of astrocytes in glutamate and GABA recycling, which then influences adjoining vessels |
| 13 | Mathias et al., 2017 [94] | Soma, dendrite, extracellular space, synaptic space, astrocyte, perivascular space, SMC, endothelial cell and lumen; I: neural activation through ion channels; O: fMRI BOLD signal | The study demonstrated the signaling method of neurovascular coupling through a model of pyramidal neurons and its analogous fMRI BOLD response. The study extended the NVU to include a complex neuron system with Na/K ATPase pump mechanism, which provides CBF and CMRO2. |

**Table 1.** (Continued)

| S. No. | Study | NVU system | Description |
|---|---|---|---|
| 14 | Kenny et al., 2018 [85] | Neuron, synaptic cleft, astrocyte, perivascular space, endothelial cell, SMC, and lumen; I: Glutamate and $K^+$ in synaptic space; O: arteriolar radius | The model used lumped parameter systems to depict the connection between a neuron and perfusing arteriole through the astrocytic perivascular $K^+$ and the SMC's $Ca^{2+}$ dynamics mediated by astrocytic EETs and TRPV4. Results indicated that $K^+$ mediated pathway drives the quick start of vaso-dilation compared to the NO-mediated pathway. |
| 15 | Mathias et al., 2018 [86] | Neuron (soma, dendrite), extracellular space, synaptic space, astrocyte, perivascular space, SMC, endothelial cell and lumen; I:neuronal current; O: fMRI BOLD signal | The model simulated NVU mechanisms and BOLD signal by extending the previous models by Mathias et al., 2017[69] and Kenny et al., 2018[59]. The study included a transient sodium ion channel in the neuron compartment. |
| 16 | Sten et al., 2020 [95] | Pyramidal neuron, GABAergic interneuron, astrocyte, SMC, arteriole; I: neuronal pulse mediating vaso-agents; O: arteriolar diameter | The study modeled the interplay between pyramidal neurons and GABAergic interneurons in the NVU. The study evaluated the role of cell-specific contributions in NVU due to the effect of an anesthetic agent. |

researchers [45,84–86] broadened the basic models by augmenting various physiological segments and detailing the NVU mechanisms.

We selected the physiologically detailed model by Witthoft and Karniadakis [45] since their study showed the bidirectional communication between cerebral astrocytes and microvessels, which was relevant due to experimental results that astrocytes are susceptible to small variations in their membrane potential [96] and their long processes are sensitive to polarization by tDCS [97–99]. In Fig 4, we compared the Witthoft and Karniadakis [45] model with two other models, Kenny et al. [85], which depicted the connection between neurons and perfusing arteriole through the astrocytic perivascular $K^+$ and the SMC's $Ca^{2+}$ dynamics, and Mathias et al. [86] model that extended Kenny et al. [85] model as explained in Table 1.

For comparison of the model responses, the physiologically detailed models were simulated using the 'ode23tb' solver in Simulink (MathWorks, Inc., USA), and the normalized vessel radius change was considered during a neuronal stimulus between 30 to 70 sec as shown in Fig 4. We found that the physiologically detailed model by Witthoft and Karniadakis [45] had a

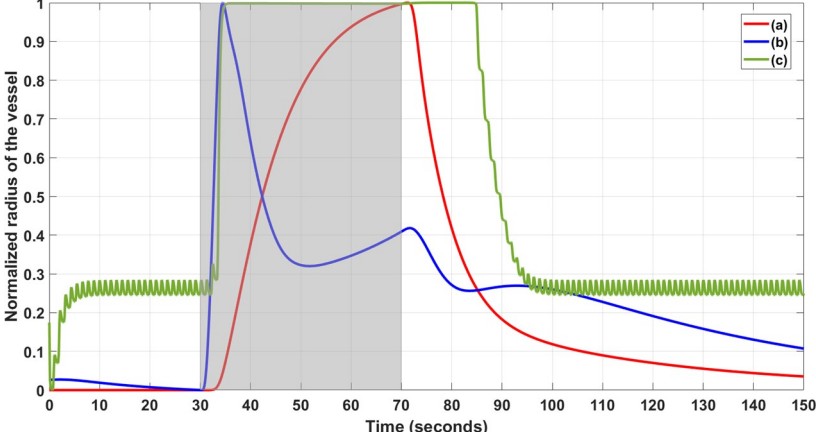

**Fig 4. Normalized vessel radius changed from the neurovascular unit models during neuronal stimulation (30 to 70 sec) shown by the shaded region.** Three relevant physiological models' simulation: Model (a) Kenny et al., 2018, Model (b) Mathias et al., 2018, and Model (c) Witthoft and Karniadakis, 2012. The period from 0 to 30 sec shows numerical transients where the model (c) settles to a non-zero vessel radius from zero initial condition. Model (c) also captured the after-effect of the neuronal stimulation on the normalized vessel radius.

**Table 2. State variables of the pathways for tDCS effects in the lumped model of the neurovascular unit for physiologically detailed modeling based on published literature [42,56,57,75].**

| Compartments of the lumped model of the neurovascular unit | | | |
|---|---|---|---|
| **Synaptic Space** | **Intracellular Astrocyte Space** | **Perivascular Space** | **Arteriole smooth muscle cell (SMC) Intracellular Space** |
| Potassium concentration in the synaptic space ($[K^+]_S$) | Astrocytic Inositol trisphosphate ($[IP_3]$) | Perivascular potassium concentration, ($[K^+]_P$) Perivascular calcium concentration ($[Ca^{2+}]_P$) | Open KIR (Inward Rectifying Potassium) channel probability ($k$) |
| | Astrocytic intracellular calcium concentration, ($[Ca^{2+}]_A$) | | SMC Membrane Potential ($V_{SMC}$) |
| | Gating variable ($h$) | | Open potassium channel probability ($n$) |
| | TRPV4 (Transient Receptor Potential Vanniloid Related 4) channel open probability ($ss$) | | Calcium concentration in the SMC ($[Ca^{2+}]_{SMC}$) |
| | Calcium-dependent EET (Epoxyeicosatrienoic Acid) production in the cell ($[EET]$) | | The fraction of attached cross-bridges ($\omega$) |
| | Open BK(Big Potassium) channel probability ($n_{bk}$) | | Normalized contractile component of length ($yy$) |
| | Astrocyte Membrane Potential ($Vk$) | | Mean circumference of the vessel ($x$) |

quick start of the vasodilation comparable to Mathias et al. [86] model that extended Kenny et al. [85] model. Also, Witthoft and Karniadakis [45] model presented with after-effect of the neuronal stimulation on the normalized vessel radius. Moreover, we found that the Witthoft and Karniadakis [45] model generated baseline vessel oscillations (see Model c in green in Fig 4) that was attenuated during activation. However, experimental investigation of non-linear limit cycle oscillations between the cerebral astrocytes and microvessels will require multi-modal imaging to measure synchronized neuronal, astrocytic calcium, and hemodynamic changes that is possible in an animal model [100]. In our human study using fNIRS, such detailed physiological model (see S1 Text) may be unidentifiable, i.e., characterized by many parameters that are poorly constrained by tHb measure, from low-density fNIRS. Therefore,

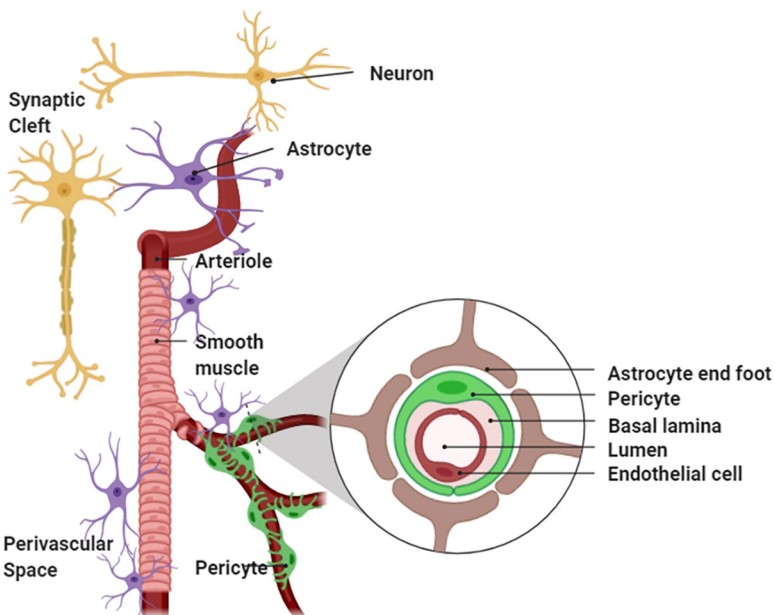

**Fig 5. Anatomical representation (created using BioRender: https://biorender.com/) of the components of neurovascular unit that can be affected by tDCS current density in the neurovascular brain tissue.**

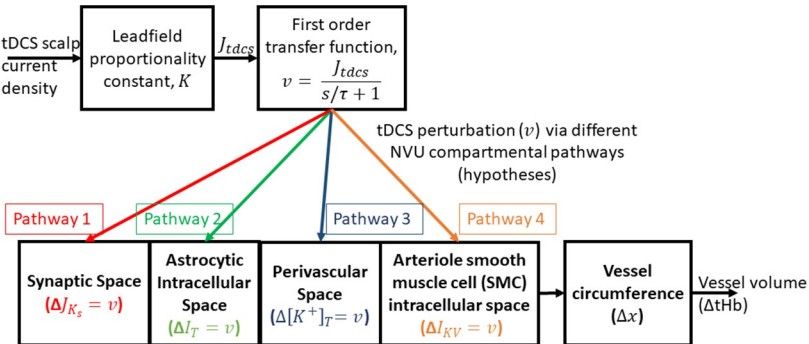

**Fig 6. Schematic representation of the four pathways for tDCS current density effects (perturbation) on vessel volume function related to vessel circumference changes for hypothesis testing using experimental total hemoglobin (tHb) concentration changes (proportional to blood volume).** The four simulated pathways are, Pathway 1: tDCS current density (input pulse) via first-order transfer function modulate vessel volume response (output) by perturbing synaptic potassium (K+) released from active neurons ($\Delta J_{K_s}$), Pathway 2: tDCS current density (input pulse) via first-order transfer function modulate vessel volume response (output) by perturbing the astrocytic transmembrane current ($\Delta I_T$), Pathway 3: tDCS current density (input pulse) via first-order transfer function modulate vessel volume response (output) by perturbing perivascular potassium (K+) concentration ($\Delta[K^+]_T$), and Pathway 4: tDCS current density (input pulse) via first-order transfer function modulate vessel volume response (output) by perturbing voltage-gated ion channel current ($\Delta I_{KV}$) on the smooth muscle cell (SMC).

we aimed to conduct hypothesis testing on tDCS effects based on initial transient blood volume changes (based on fNIRS tHb) such that the reduced dimension grey-box model is parsimonious (lower model order better) and also conform to fNIRS tHb data (goodness of fit). Here, reduced dimension grey-box modeling is crucial since under-fitting is known to induce bias while over-fitting induces variability. So, Akaike information criterion (AIC) was used to find the balance where tDCS was postulated to affect various NVU compartmental states, i.e., our hypotheses for model selection [101], that are presented in the next section.

## (D) Physiologically detailed neurovascular compartmental dynamics

A physiologically detailed mathematical model of tDCS effects on the compartments of the lumped model of the NVU (state variables tabulated in Table 2) was simulated based on published literature (see Table 1). The anatomical representations are shown in Fig 5 where besides neurons, the major non-neuronal glial cells in the brain, astrocytes, are also susceptible to small variations in their membrane potential [96], and their long processes are sensitive to polarization by tDCS [97–99]. In animal studies, tDCS has been found to induce astrocytic calcium waves in the visual cortex to steer plasticity of the visually evoked potentials [102]. Further, electrically coupled populations of glial cells, known as a glial syncytium, can intensify field polarization in response to tDCS. Likewise, vascular cells such as endothelial cells and arteriolar SMC can be affected by tDCS through the modulation of nitric oxide (NO) and neuropeptide Y (NPY), perivascular neurotransmitters, and polarization of SMC causing vasomotion via metabolites such as potassium ions (K+), adenosine, NO and calcium ions (Ca$^{2+}$) [103]. Our grey-box model was developed from the physiologically detailed models by Witthoft and Karniadakis [45] that generated rapid response, vessel oscillations, and stimulation aftereffects.

Neuronal effects due to tDCS current density can take an intricate path to the synaptic transmembrane current, considering only excitatory effects that can be mapped through a sigmoid function as presented by Molaee-Ardekani et al. [104]. In this current study, we did not explicitly model neuronal dynamics, so we did not capture neuroplastic changes [17] and

neuronal excitability after-effects induced by longer duration (>3 min) tDCS. To capture the immediate transient effect of tDCS, the major signaling pathways were considered in our model, viz., neural synaptic potassium (K+) and glutamate to astrocytes, K+ signaling between astrocytes and microvasculature, and microvasculature to astrocytes interactions via astrocyte perivascular endfoot [45]. The detailed differential equations and model parameters are provided in the S1 Text. Our goal was mechanistic grey-box modeling for hypothesis testing, where the hypotheses are formulated as a set of mathematical equations for data fitting to identify core predictions from biological criteria [105]. We identified four nested NVU compartmental pathways (see Fig 6) where tDCS perturbed a state variable at each of the four NVU compartments (see Table 2 for details on compartmental state variables) for hypothesis testing of the initial (0-150sec) tDCS effects on the NVU leading to blood vessel (circumference) volume changes. Fig 6 shows the nested model for each of the four pathways starting from tDCS scalp current density that perturbed synaptic potassium (K+) released from active neurons ($\Delta J_{K_s}$) for Pathway 1, astrocytic transmembrane current ($\Delta I_T$) for Pathway 2, perivascular potassium (K+) concentration ($\Delta [K^+]_T$) for Pathway 3, and voltage-gated ion channel current ($\Delta I_{KV}$) on the smooth muscle cell (SMC) for Pathway 4.

For NVU compartmental modelling of the tDCS perturbations (see Fig 6), the tDCS current density in the brain's neurovascular tissue was assumed to be proportional to the tDCS current density applied at the scalp due to the Ohmic volume conductor head model (see S2 Text). Prior works have shown that the change in the concentration of various vasoactive agents can be represented as a vasoactive signal with first-order Friston's model [106]. So, the tDCS current density at the scalp ($I_{tdcs}$) was proportional to the current density ($J_{tDCS}$) in the neurovascular brain tissue based on a lead field matrix (the forward solution; see S2 Text) leading to the vasoactive signal (state perturbation to NVU compartments) via first order transfer function,

$$v_i = \frac{K_i}{s/\tau + 1} I_{tdcs} \tag{i}$$

where $K_i$ is arbitrary gain from lead field matrix ($J_{tDCS} = K_i \cdot I_{tdcs}$), and $\tau$ is the time constant. Here, the state variables of various NVU compartments modeled in the study are listed in Table 2, while the detailed equations are provided in the S1 Text.

**(a) Pathway 1:** tDCS perturbation of synaptic potassium leading to vessel circumference changes

Studies have shown that K+ can act as a potent vasodilator signal that couples local neuronal activity to vasodilation in the brain, and have a major role in cerebrovascular mechanisms [107–110]. Studies have shown that the potassium pathway is responsible for the fast onset of vasodilation compared to the other mediators [85,86,111]. Here, synaptic activity was assumed to be modulated by tDCS current density [112] that affected K+ release from active neurons into the synaptic space. Here, $J_{K_s}$ is the K+ released from active neurons that is considered to be perturbed by tDCS from its baseline condition such that $\Delta J_{K_s} = v_1 = \frac{K_1}{s/\tau+1} I_{tdcs}$ using Eq (i). The potassium concentration in the synaptic space, $[K^+]_s$ is then given as (see Eq 1 in the S1 Text):

$$\frac{d[K^+]_s}{dt} = J_{K_s} + \frac{K_1}{s/\tau + 1} I_{tdcs} - J_{\Sigma Kmax} k_{Na} \frac{[K^+]_s}{[K^+]_s + KKO_a} \tag{ii}$$

where $J_{K_s}$ is the baseline flux of K+ in the synaptic space, $J_{\Sigma Kmax}$ is the maximum flux, $k_{Na}$ is a constant parameter that depends on extracellular sodium concentration, $KKO_a$ is the threshold value for K+ concentration in the synaptic space, $[K^+]_s$.

**(b) Pathway 2:** tDCS perturbation of the astrocytic transmembrane current leading to vessel circumference changes

Astrocytes are susceptible to small variations in their membrane potential [96] and their long processes are sensitive to polarization by tDCS [97–99]. Here, astrocytic transmembrane current ($I_T$) was perturbed by tDCS, $\Delta I_T = v_2 = \frac{K_2}{s/\tau+1} I_{tdcs}$ from Eq (i) that was added to other transmembrane currents including $I_{BK}$ is current through Big Potassium (BK) channel, $I_{leak}$ is leak current, $I_{TRPV}$ is electrical current through the TRPV channel and $I_{\Sigma K}$ is the electrical current carried by the K$^+$ influx at the perisynaptic process (Eq 21 in the S1 Text):

$$\frac{dV_k}{dt} = \frac{1}{C_{astr}} \left( -I_{BK} - I_{leak} - I_{TRPV} - I_{\Sigma K} + \frac{K_2}{s/\tau+1} I_{tdcs} \right) \qquad \text{(iii)}$$

**(c) Pathway 3:** tDCS perturbation of perivascular potassium concentration leading to vessel circumference changes

Glial cells maintain extracellular K+ concentration by the imbalance in their membrane polarity and can affect K+ spatial buffering affecting tDCS modulation [113] of neurovascular coupling [35]. Astrocytic release of K+, via two potassium channels (BK and KIR), into the perivascular space can be perturbed by tDCS. Astrocytic role in neurovascular coupling may be related to the strength of stimulation where high strength can also lead to vasoconstriction [100] (mediated via K+ and EET signaling [83]). Vasoconstriction can also follow vasodilation when the astrocytic calcium concentration (or, perivascular K+ concentration) increase above a certain threshold. We assumed that low-intensity tDCS perturbation would not cross that threshold where $\Delta[K^+]_T = v_3 = \frac{K_3}{s/\tau+1} I_{tdcs}$. So, the perivascular potassium concentration, $[K^+]_P$, is given as (Eq 26, S1 Text):

$$\frac{d[K^+]_P}{dt} = \frac{J_{BK}}{VR_{pa}} + \frac{J_{KIR}}{VR_{ps}} - R_{decay}\left([K^+]_P - [K^+]_{P,min}\right) + \frac{K_3}{s/\tau+1} I_{tdcs} \qquad \text{(iv)}$$

Here, $[K^+]_{P,min}$ is the resting state equilibrium K$^+$ concentration in the perivascular space. The K+ flow from the astrocyte and SMC are $J_{BK}$ and $J_{KIR}$ corresponding to BK and inward rectifying potassium (KIR) respectively. And, $VR_{pa}$ and $VR_{ps}$ are the volume ratios of perivascular space to astrocyte and SMC, respectively. $R_{decay}$ is the rate at which perivascular K$^+$ concentration decays to its baseline state.

**(d) Pathway 4:** tDCS perturbation of voltage-gated ion channel current on the smooth muscle cell leading to vessel circumference changes.

Smooth muscle cell (SMC) voltage-gated potassium (KV) channels and inwardly rectifying K+ channels are important in penetrating arterioles that control arterial diameter by exerting a major hyperpolarizing influence [114]. Therefore, tDCS electric field can perturb voltage gated potassium current ($\Delta I_{KV} = v_4 = \frac{K_4}{s/\tau+1} I_{tdcs}$) that was added to other currents including $I_L$, $I_K$, $I_{Ca}$ and $I_{KIR}$ that represent leak, K+, Ca2+, and KIR channel currents respectively in the SMC compartment. Then, the SMC membrane potential, $V_{SMC}$, is given by (Eq 41, see S1 Text):

$$\frac{dV_{SMC}}{dt} = \frac{1}{C_{SMC}} \left( -I_L - I_K - I_{Ca} - I_{KIR} - I_{KV} + \frac{K_4}{s/\tau+1} I_{tdcs} \right) \qquad \text{(v)}$$

Here, the four tDCS perturbation pathways are nested, i.e., the pathway 1 is represented by seventeen ordinary differential equations starting from synaptic K+ (see S1 Text and S1 Table) that nested other tDCS perturbation pathways 2–4 –see Fig 6.

### (E) Physiologically detailed model linearization for grey-box analysis of fNIRS data:

In our current study, a grey-box linear model was developed from a detailed physiological model to analyze tDCS-evoked tHb changes for hypothesis testing. Although the detailed non-linear model can be fitted using advanced methods, e.g., simulated annealing (Optimization Toolbox—MATLAB–MathWorks, USA); however, simplifying the model is necessary to identify key mechanisms of the system and to understand relevant aspects as shown as pathways in Fig 6. Therefore, we applied grey-box linear modelling where many states and parameters were removed for the identification of the core predictions based on our biological criteria [105]. For the grey-box analysis of the fNIRS-tHb (normalized) as output to the input tDCS current waveform (normalized), model reduction of the four pathways from the physiologically detailed model was first performed using Simulink's linear analysis tool (MathWorks, Inc., USA). Model Linearizer tool allows linearization of complex nonlinear models at different operating points. This tool allowed the linearization of complex physiologically detailed NVU model at their baseline operating point found from published literature (see S1 Table). We assumed that the subthreshold tDCS current density perturbation to the different pathways of the lumped NVU model would operate close to the resting-state baseline conditions at the NVU compartments (blocks in Simulink model), during the initial transient period (<3 min) after tDCS onset, until neuroplastic changes occur (neuroplastic after-effects >3 min of tDCS [19]). Therefore, the physiologically detailed model linearization was performed using a block-by-block approach (i.e., NVU compartment-by-compartment) at the initial conditions (from published literature, see S1 Table) such that the Model Linearizer tool (MathWorks, Inc., USA) individually linearized each block (or, NVU compartment) in the physiologically detailed Simulink model and produced the linearization of the overall NVU system by combining the individual block linearization. Here, the linearization step approximated the system of nonlinear differential equations around the baseline resting-state conditions for each NVU compartment. The resultant linear model presented the NVU system as a set of input, internal states (compartment variables–see Table 2), and output as transfer functions, which depicted the relationship between input tDCS current waveform (normalized) and the output tHb (normalized) hemodynamic response. Therefore, the linearized grey-box model was constrained by the physiology of the respective four tDCS perturbation pathways (see Fig 6) and compartments of the physiologically detailed NVU model (see Table 2). The linearized model was then used for grey-box linear modeling with identifiable parameters ('idgrey' in MATLAB, MathWorks, Inc., USA).

### (F) Subject-specific Grey-box Linear Model Dynamics from individual fNIRS data:

Grey-box linear model of four physiologically detailed tDCS perturbation pathways was found using Model Linearizer tool in Simulink (MathWorks, Inc., USA) linear analysis package as described earlier. Then, the grey-box models were evaluated using experimental fNIRS data fitting based on the cost function that sums the squared and normalized residuals. The lumped model of the NVU assumed a system of input (tDCS perturbation at NVU compartments–see Fig 6) and outputs (change in terms of vessel circumference). The dynamics of NVU were considered to capture the effects of tDCS on CVR through signaling mechanisms across four compartments, as shown in Fig 6. For modeling the output vessel function and hemodynamics, a cylindrical vessel component having a unit length was considered as a lumped model of the blood vessel [115]. Here, the blood volume changes were assumed to be proportional to the vessel circumference under the assumption of small circumference changes. Then, the changes

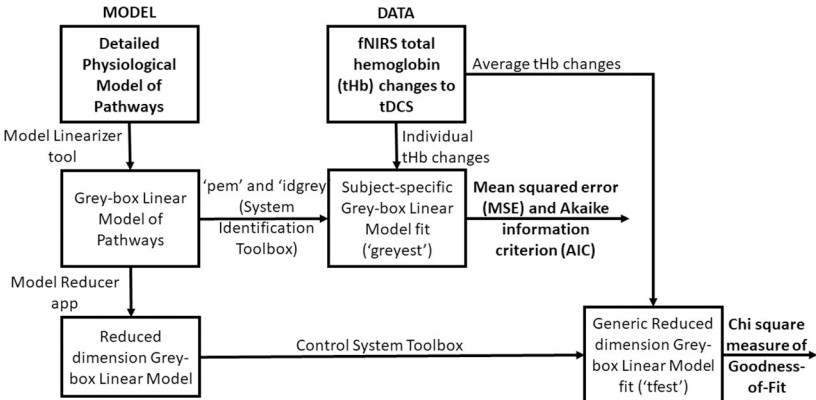

**Fig 7. Grey-box model estimation from MODEL and DATA using System Identification Toolbox (Mathworks Inc., USA) for hypothesis testing based on mean squared error, Akaike information criterion, and Chi-square Goodness-of-fit.**

in tHb were considered proportional to the blood volume [72], so lumped volume response using vessel circumference was linearly mapped to the tHb changes under small change approximation.

Fig 7 shows our model estimation work flow diagram using the System Identification Toolbox (Mathworks, Inc., USA). The grey-box linear model with identifiable free parameters ('idgrey' function) was updated using the "Refine Existing Model" approach ('greyest' function for 'idgrey' model) in the System Identification Toolbox (Mathworks, Inc., USA) that uses prediction-error minimization (PEM) algorithm ('pem' function) to update the parameters of an initial model (from Model Linearizer tool–results in S3 Table) to improve the fit to the estimation fNIRS data of each subject–so subject-specific grey-box linear modelling. PEM technique considered the accuracy of the predictions computed for the observations and most tightly connected to systems theory as it explicitly exploits the dynamical structure of the studied system [116]. So, the subject-specific model evaluation in PEM was based on the properties of the prediction-error cost function for each pathway (i.e., hypothesis) for each subject constrained by the physiology of the respective tDCS perturbation pathways (see S1 Text and S1 Table). We computed subject-specific MSE and AIC for each pathway (i.e., hypothesis) for grey-box linear model fitting.

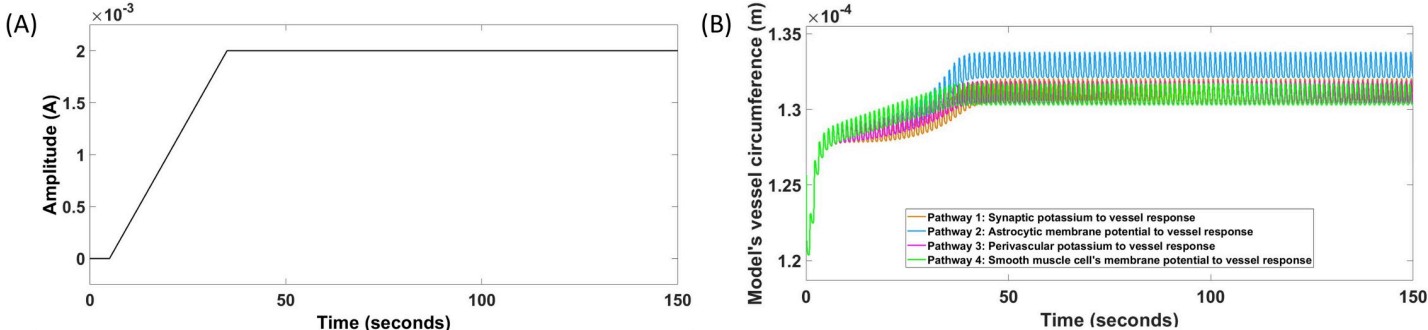

**Fig 8.** Physiologically detailed neurovascular coupling model showing the input HD-tDCS current (A) and the output vessel circumference response for the four simulated pathways for duration 0-150sec (B). Pathway 1: tDCS current density perturbing vessel circumference through synaptic potassium pathway, Pathway 2: tDCS current density perturbing vessel circumference through the astrocytic pathway, Pathway 3: tDCS current density perturbing vessel circumference through perivascular potassium pathway, and Pathway 4: tDCS current density perturbing vessel circumference through smooth muscle cell voltage-gated ion current channels pathway. Shown in (B), all the tDCS perturbation pathways has an initial transient response, and then generated steady-state vessel oscillations.

### (G) Investigation of a Reduced Dimension Grey-box Linear Model for averaged fNIRS data:

Linearized grey-box model's complexity (see S3 Table) was simplified with reduced-order approximations using the Model Reducer app in the Control System Toolbox (Mathworks, Inc., USA). Simpler models can preserve model characteristics while discarding states that contribute relatively little to system dynamics (Balanced Truncation Model Reduction, 'balred' function). Simpler transfer function models can provide insights into the linear time-invariant model dynamics that were derived from the minimal realization transfer function for the four tDCS perturbation pathways. A generic reduced dimension grey-box linear model was derived ('tfest' function) using the nested linearized grey-box models for the initial parameterization then fitted to averaged fNIRS-tHb data–see Fig 7. We used Chi-Square Goodness-of-Fit for comparing the four hypotheses for the nested pathways (see Fig 6) where Chi-Square difference test for nested pathways [117] determined the tDCS perturbation pathway model of the least order.

## Results

### (A) Physiologically detailed neurovascular compartmental dynamics:

A physiologically detailed neurovascular coupling model was developed from published literature (see S1 Table), specifically from physiologically detailed model by Witthoft and Karniadakis [45], as described in the Methods section (differential equations and model parameters in the S1 Text and S1 Table). The four simulated tDCS perturbation pathways are shown in Fig 8; 1) tDCS perturbing vessel response through synaptic potassium pathway, 2) tDCS perturbing vessel response through the astrocytic pathway, 3) tDCS perturbing vessel response through perivascular potassium pathway, and 4) tDCS perturbing vessel response through SMC voltage-gated ion channels pathway. Fig 8A shows the HD-tDCS input that is a trapezoidal current waveform and Fig 8B shows the simulated model output that is the blood vessel circumference, $x$. Here, 2mA tDCS trapezoidal waveform, shown in Fig 8A, generated a state perturbation (see Eqs ii–v) in the four NVU compartments of the lumped model (see Fig 6) after first-order filtering (Eq i; 20ms passive membrane time constant [118]). Fig 8B shows that all the four tDCS perturbation pathways of the physiologically detailed Witthoft and Karniadakis [45] model generated steady state vessel oscillations for subthreshold stimulation. This is because all the four pathways are nested (see Fig 6) where the last pathway 4 leads to the vessel oscillations.

### (B) Physiologically detailed model linearization for grey-box analysis using subject-specific fNIRS-tHb data:

We limited the fNIRS-tHb data within the lower frequency band (0.01–0.05Hz) to capture the initial transient response (not steady-state vessel oscillations) [29] during first 150sec (30sec ramp-up+120sec steady-state) of tDCS perturbation. System Identification Toolbox (MathWorks, Inc., USA) was used for grey-box modelling of the time-domain tDCS input (current density waveform) and fNIRS output (normalized tHb) data from eleven healthy subjects. The input time-series was a normalized trapezoidal waveform, and the output time-series were normalized values of the subject-specific fNIRS-tHb changes from baseline during the initial 150 secs (30sec ramp-up and 2 min steady-state so < 3 min) of HD-tDCS at the targeted-region and the nontargeted-region of the bilateral sensorimotor cortex (montage is shown in Fig 1). The grey-box linear model with identifiable parameters obtained after physiologically detailed model linearization was taken as the initial system to fit the

experimental fNIRS-tHb data from each subject. Initial 150 sec (ramp-up period of 30 sec + 120 sec steady-state) was assumed to be unaffected by neuroplastic effects of tDCS [19]. Fig 9 shows the simulated grey-box linear model output, i.e., normalized fNIRS-tHb (proportional to vessel volume change) of the four pathways for each of the 11 subjects, P1-P11 (individual grey-box transfer functions are tabulated in the S3 Table). Fig 9A1–9A4 shows the four simulated pathways at the targeted-region while Fig 9B1–9B4 shows the four simulated pathways at the nontargeted-region of the bilateral sensorimotor cortex for each subject. Fig 9 also shows the ensemble averaged fNIRS-tHb response data for 0-150sec across all subjects with a dashed line (from Fig 3B and 3C). Then, Fig 10 shows the boxplot of the mean square error (MSE) and the Akaike information criterion (AIC) across 11 subjects from subject-specific grey-box analysis after physiologically detailed model linearization. Notches display the variability of the median between samples. The width of a notch is computed so that the boxes whose notches do not overlap have different medians at the 5% significance level. Here, for MSE of the four pathways for the targeted-region, the mean are 0.029, 0.065, 0.024, 0.042, the standard deviation are 0.020, 0.062, 0.020, 0.033, and the median are 0.023, 0.054, 0.018, 0.033 respectively. Then, the MSE of the four pathways for the nontargeted-region, the means are 0.052, 0.123, 0.118, 0.058, the standard deviation are 0.032, 0.113, 0.096, 0.056, and the median are 0.043, 0.068, 0.086, 0.031 –see S5 Table. Then, for AIC of the four pathways for the targeted-region, the means are -0.459, -0.216, -1.743, -1.748, the standard deviation are 0.862, 1.496, 1.433, 1.173, and the median are -0.606, -0.162, -1.726, -1.659 –see S6 Table. Then, for AIC of the four pathways for the nontargeted-region, the means are 1.386, 1.313, 0.478, -0.747, the standard deviation are 0.599, 1.264, 1.346, 0.966, and the median are 1.366, 1.261, 0.814, -0.915. Fig 10A shows that the tDCS perturbation Pathway 3, from the perivascular K+ to vessel circumference, presented the least MSE (median <2.5%) across all subjects at the targeted-region, followed by the Pathway 1. Fig 10B shows that the tDCS perturbation Pathway 4 gave lowest median MSE and medial AIC across all subjects for the grey-box linear model fits at the contralateral nontargeted-region. Here, over-fitting induces variability while under-fitting is known to induce bias so we used Akaike information criterion (AIC) to find a balance (lowest AIC selected). Fig 10C shows that the tDCS perturbation Pathway 3 presented the lowest AIC (median -1.726) across all subjects for the grey-box linear model fits at the targeted-region while Fig 10D shows that the tDCS perturbation Pathway 4 gave the lowest AIC (median -0.915) across all subjects at the contralateral nontargeted-region. Residuals checks were performed based on the autocorrelation curves of the residuals and the cross-correlation curves between input and the residuals–see S3 Fig. Therefore, based on MSE and AIC, tDCS perturbation Pathway 3 was selected for the grey-box linear model fits at the targeted-region while Pathway 4 was selected for the contralateral nontargeted-region. For MSE comparison with long-term (0-600sec) tDCS effects, grey-box modeling of the complete 10 min of tHb data at the targeted-region was performed (see S1 Fig) where the perturbation Pathway 3 again resulted in the lowest MSE (median <0.5%)–see S2 Fig.

## (C) Reduced dimension grey-box linear model analysis of averaged fNIRS-tHb data:

The average tHb time series during initial transient 150 sec of HD-tDCS from eight subjects with anti-correlated oxy-Hb & deoxy-Hb for 600 sec (i.e., correlation coefficient <-0.5 between oxy-Hb & deoxy-Hb at targeted-region–see S2 Table) was used for model fitting of the minimal realization transfer function. Fig 11 shows the subjects P3, P4, and P10 who were rejected due to poor anti-correlated oxy-Hb & deoxy-Hb. Fig 11 also shows an initial dip (0-150sec) in the tHb in these subjects that is also present in oxy-Hb & deoxy-Hb timeseries.

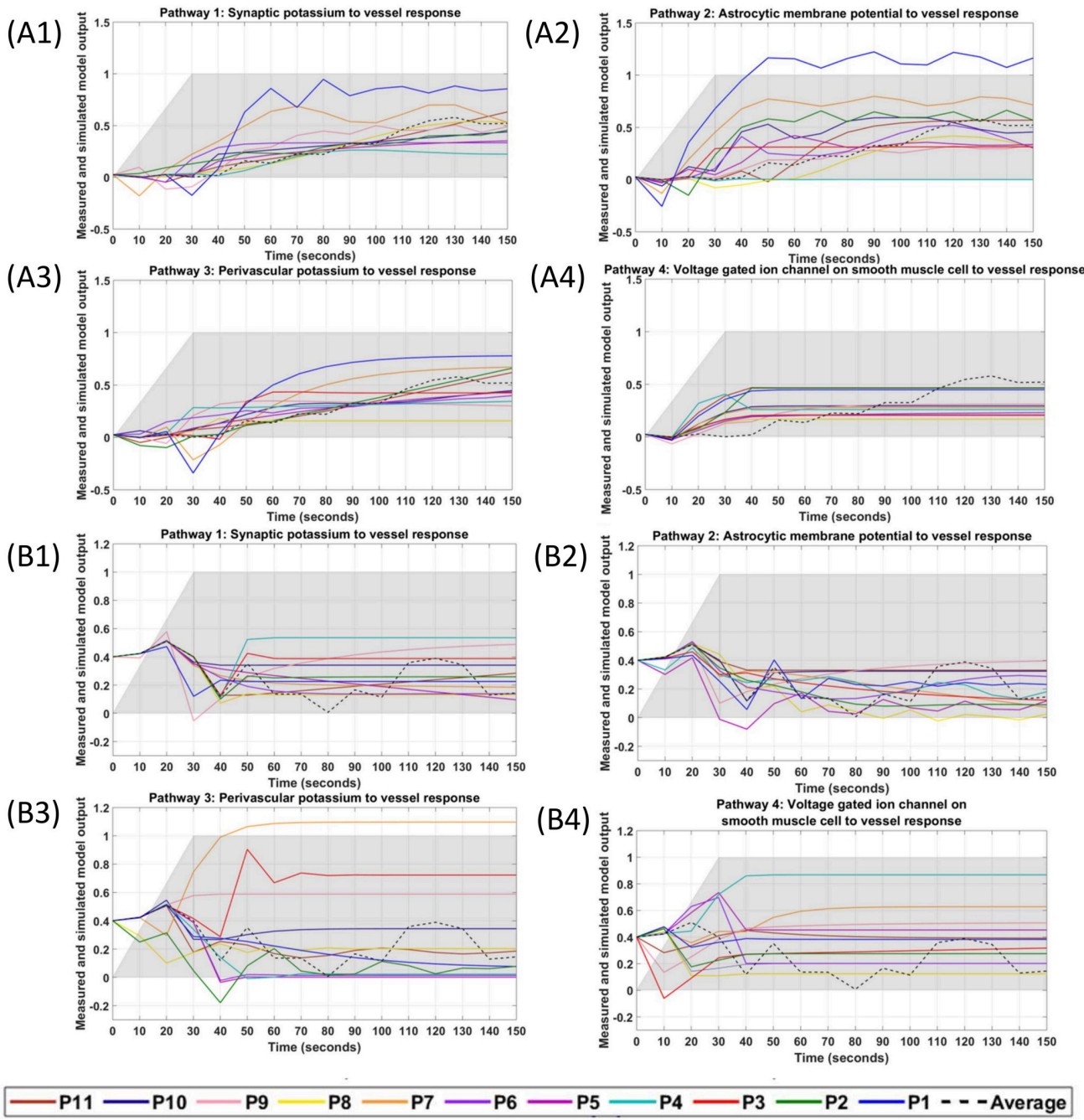

**Fig 9. Simulated grey-box linear model output for the four pathways fitted individually to each of the 11 subjects' fNIRS-tHb data during HD-tDCS (shaded grey).** The ensemble averaged fNIRS-tHb response across all subjects is shown with a dashed line. Plots (A1)–(A4) show the four pathways fitted to the HD-tDCS stimulated sensorimotor data, while the plots (B1)–(B4) show the four pathways fitted to the HD-tDCS non-targeted region of the contralateral sensorimotor cortex. Pathway 1: tDCS current density perturbing vessel (circumference) response through synaptic potassium pathway, Pathway 2: tDCS current density perturbing vessel (circumference) response through the astrocytic pathway, Pathway 3: tDCS current density perturbing vessel (circumference) response through perivascular potassium pathway, and Pathway 4: tDCS current density perturbing vessel (circumference) response through smooth muscle cells voltage-gated ion current channels pathway.

Table 3 shows the reduced-order approximations of high-order grey-box linear models (see S3 Table). All the tDCS perturbation pathways have an excess of model poles over the number of zeros; therefore, the frequency response magnitude will tend to zero with an increasing

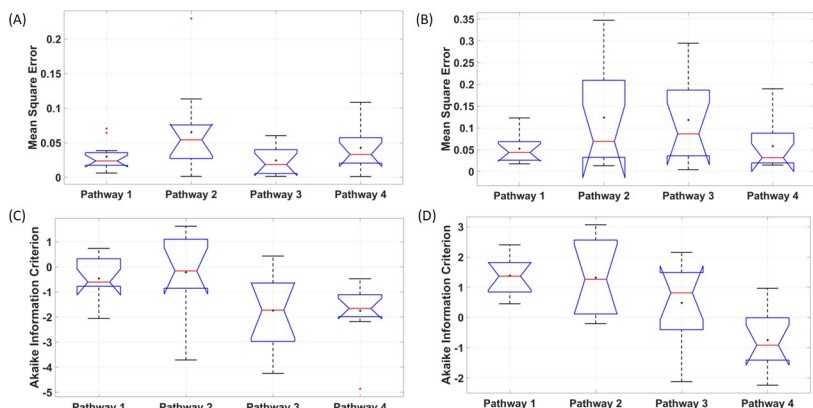

**Fig 10. Boxplot across 11 subjects of the mean square error (MSE) and Akaike Information Criterion (AIC) of grey-box linear model fits for the four tDCS perturbation pathways.** Pathway 1: tDCS perturbing vessel response through synaptic potassium pathway, Pathway 2: tDCS perturbing vessel response through the astrocytic pathway, Pathway 3: tDCS perturbing vessel response through perivascular potassium pathway, and Pathway 4: tDCS perturbing vessel response through the SMC pathway. On each box, the central mark indicates the median, and the bottom and top edges of the box indicate the 25th and 75th percentiles, respectively. If the notches in the box plot do not overlap, one can conclude, with 95% confidence, that the true medians do differ. The whiskers extend to the most extreme data points not considered outliers, and the outliers are plotted individually using the red '+' symbol. The mean is also shown with a black '+' symbol. (A) MSE for targeted-region. (B) MSE for nontargeted-region. (C) AIC for targeted-region. (D) AIC for nontargeted-region.

frequency. However, the numbers of model zeros are different in the four pathways, where the positive real zeros are most likely approximating the system's time delay. All four pathways have complex conjugate poles (from $s^2 + 9.804s + 95.24$ terms) in the stable region. Fig 12A shows the normalized impulse response function of the minimal realization transfer function for the four pathways where Pathway 1 (TF1(s)) had 11 poles, and 3 zeros, Pathway 2 (TF2(s)) had 10 poles, and 3 zeros, Pathway 3 (TF3(s)) had 8 poles and 2 zeros. Pathway 4 (TF14(s)) had 6 poles and 1 zero as tabulated in Table 3.

Minimal realization transfer functions provided a qualitative analysis for CVR where the tDCS perturbation Pathway 4 had the fastest response (peaked at 0.4 sec), and the tDCS perturbation Pathway 1 had the slowest response (peaked at 5 sec) as shown in Fig 12A–as expected based on their nested hierarchy (see Fig 6). Pathway 1, acting via the K+ released from active neurons in to the synaptic space, resulted in the stereotypical time-to-peak [119] in the hemodynamic response function of about 5 seconds after stimulus onset [120], as shown in Fig 12A. Fig 12B shows that none of the minimal realization transfer functions had an initial dip since subjects P3, P4, P10 with initial dip in fNIRS-tHb were rejected before reduced dimension linear model analysis. After cascading with the first-order transfer function for tDCS waveform (see Eq i), the four pathways' minimal realization transfer functions (Table 3) were found from the average tHb time series across eight remaining subjects. Fitting of the least complex (model order) Pathway 4 model, TF4, with 7 poles and 1 zero (all free parameters) provided an MSE of 0.0031 and Chi-Square Goodness of Fit of 0.0104. Then, the next complex (model order) Pathway 3 model, TF3, with 9 poles and 2 zeros (all free parameters) provided an MSE of 0.0025 and Chi-Square Goodness of Fit of 0.0078. Then, the next complex (model order) Pathway 2 model, TF2, with 11 poles and 3 zeros (all free parameters) provided an MSE of 0.0025 and Chi-Square Goodness of Fit of 0.0085. The most complex (model order) Pathway 1 model, TF1, with 12 poles and 3 zeros (all free parameters) provided an MSE of 0.0264 and Chi-Square Goodness of Fit of 0.0647. Here, more parameters in more complex (model order) models worsened the Chi-Square Goodness of Fit even with similar MSE, e.g.,

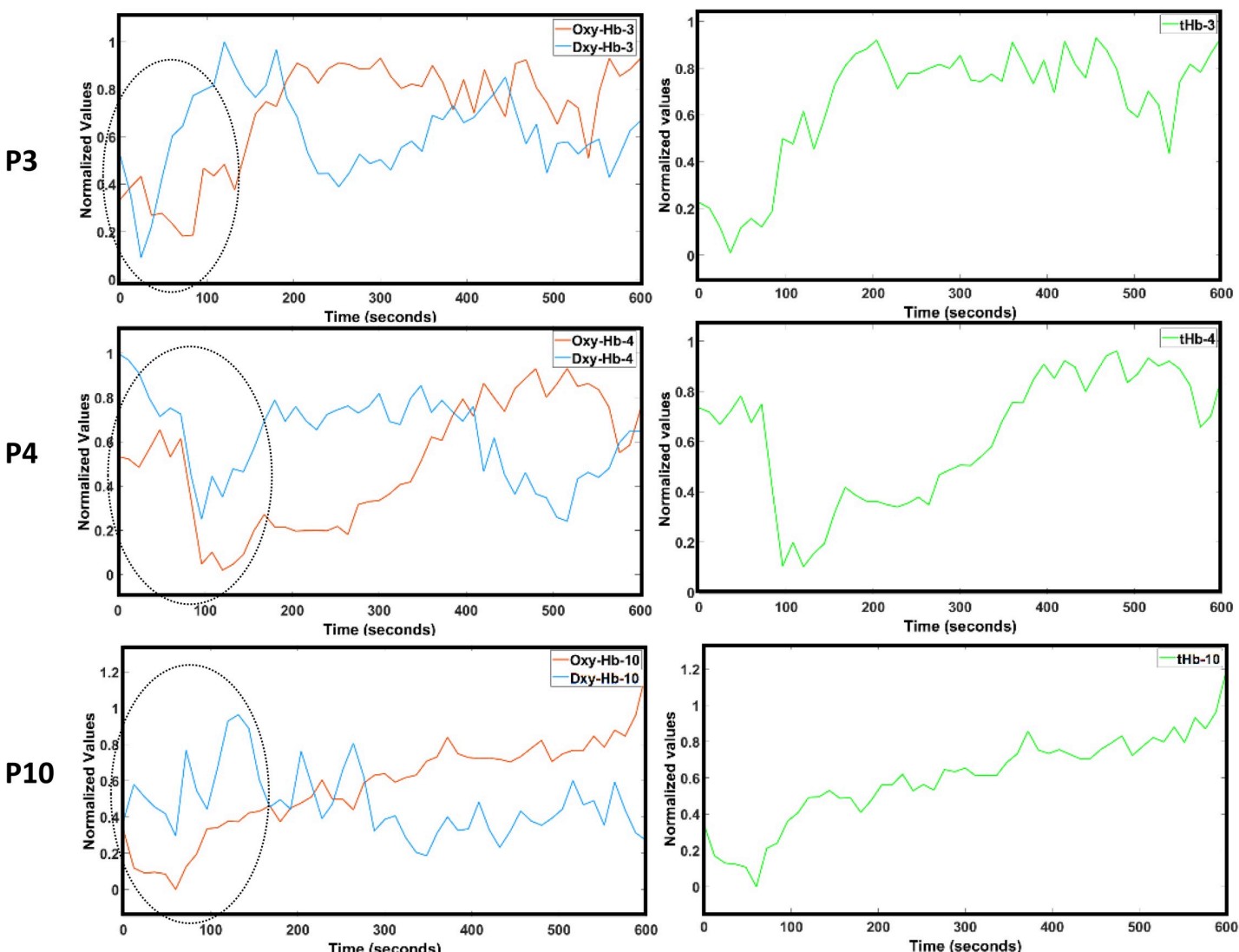

**Fig 11. Oxy-hemoglobin (Oxy-Hb) timeseries in red, deoxy-hemoglobin (Dxy-Hb) timeseries in blue in the left panels, and total-hemoglobin (tHb) timeseries in green in the right panels at the targeted-region for 0-600sec during anodal HD-tDCS for the subjects P3, P4, P10.** These subjects were rejected due to poor (correlation coefficient >-0.5) anti-correlated Oxy-Hb & Dxy-Hb measures of -0.0692, -0.1900, and -0.3768 respectively. These subjects also demonstrated an initial dip in the tHb (also, in Oxy-Hb & Dxy-Hb) timeseries that is highlighted with a black ellipse in the left panels for the Oxy-Hb & Dxy-Hb timeseries.

TF2 higher model order than TF3. Therefore, TF3 for the tDCS perturbation Pathway 3 provided a balance in terms of Chi-Square difference test for nested pathways [117] for the HD-tDCS-induced initial transient tHb changes within 0.01 and 0.05 Hz.

## Discussion

We performed a grey-box linear systems analysis of the fNIRS-based CVR based on the changes in the tHb to anodal HD-tDCS perturbation in healthy humans. Our study on fNIRS-based CVR to tDCS is supported by prior works that have evaluated the hemodynamic effects of tDCS using fNIRS in humans. Merzagora et al. [121] assessed the changes in prefrontal cortical oxygenation related to tDCS using fNIRS in healthy subjects at rest. A large increase in oxy-Hb was observed in the 10min period following anodal tDCS compared to baseline levels

**Table 3. Minimal realization transfer functions were obtained through Model Reducer (MATLAB, MathWorks, Inc., USA) for four model pathways from the detailed physiological model for the neurovascular unit compartment model.**

| Pathway transfer functions (TF) | Transfer functions from the compartmental neurovascular coupling model that were used for the initial parameterization of the four pathways | Parameterization |
|---|---|---|
| **TF1.** synaptic potassium → vessel circumference | $TF1(s) = \frac{1}{(s+0.4)} TF2(s)$ | 11 poles, 3 zeros |
| **TF2.** astrocytic membrane potential → vessel circumference | $TF2(s) = \frac{(s+46.5)}{(s+1.966)(s+15.08)} TF3(s)$ | 10 poles, 3 zeros |
| **TF3.** perivascular potassium → vessel circumference | $TF3(s) = \frac{(s+2.371e07)}{(s+2.974e04)(s+1)} TF4(s)$ | 8 poles, 2 zeros |
| **TF4.** SMC voltage gated ion channel → vessel circumference | $TF4(s) = \frac{(s+2.962)}{(s+9.594e06)(s+20.69)(s+3.3)(s+0.2446)(s2+9.804s+95.24)}$ | 6 poles, 1 zero |

before tDCS. Muthalib et al [52] showed that anodal HD-tDCS induced increases in oxy-Hb during 10min of stimulation (at 2mA) in the sensorimotor cortex region within the vicinity of the 4 x 1 HD-tDCS montage compared to the region outside this boundary. At the same time, there were minimal oxy-Hb changes in the contralateral non-targeted sensorimotor cortex region. Yaqub et al. [122] evaluated the prefrontal cortex resting-sate intra-hemispheric and inter-hemispheric connectivity changes induced by 10min (1mA) anodal 4x1 HD-tDCS in healthy subjects. Compared to the pre-stimulation phase, Yaqub et al. [122] observed that the oxy-Hb levels and the resting-state connectivity of the prefrontal cortex increased during and after the stimulation, and the connectivity changes were more in the vicinity of the 4x1 HD-tDCS electrodes. Sood et al. [50] extended Dutta et al. [36] and presented an autoregressive model parameter estimator method using Kalman filter to evaluate the relationship between changes in the fNIRS oxy-Hb signal and the EEG bandpower signal during anodal HD-tDCS. The time-varying poles of the autoregressive model were found to be comparable in all the healthy subjects. In contrast, the zeros of the model exhibited variations across the subjects

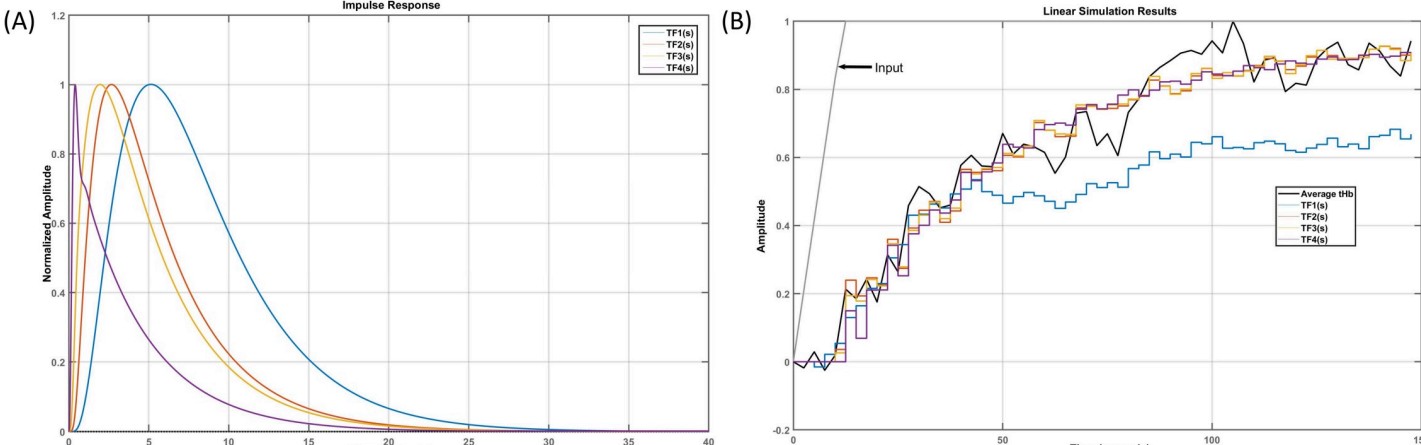

**Fig 12.** (A) Normalized impulse response function of the minimal realization transfer function (see Table 3) for the four pathways (TF1(s): tDCS perturbation of the vessel response through synaptic potassium pathway, TF2(s): tDCS perturbation of the vessel response through the astrocytic pathway, TF3(s): tDCS perturbation of the vessel response through perivascular potassium pathway, and TF4(s): tDCS perturbation of the vessel response through smooth muscle cell pathway) of the reduced dimension grey-box linear. (B) Linear model simulation of the minimal realization transfer functions (TF1-TF4) of the four pathways fitted (all free parameters) to the averaged (across subjects) normalized tHb time series (0–150sec).

during HD-tDCS that indicated modulation of the neurovascular coupling [35]. Here, simultaneous monitoring of the hemodynamic response is crucial for dosing tDCS due to its postulated role in neuromodulation action [8], which can lead to inter- and intra- subject variability in neuronal responses [49,123]–need for closed-loop dosing [124]. Furthermore, tDCS can be a promising method to evoke regional CBF [5] to ameliorate hypoperfusion in cerebrovascular diseases, including facilitating cognitive rehabilitation.

The hemodynamic response to tDCS current density in the brain can be captured based on the tHb changes using our parameterized grey-box linear model. In the current study, fNIRS data consisted of changes in tHb at the ipsilateral (to HD-tDCS) targeted-region and the contralateral nontargeted-region sensorimotor cortex, where the biological criteria was formulated in the neurovascular coupling frequency band of 0.01–0.05 Hz for investigation. We applied a system identification approach using a physiologically constrained linear model to capture the fNIRS-based CVR to anodal HD-tDCS in healthy humans where the pathway from the perivascular K+ to vessel circumference (i.e., Pathway 3) presented the lowest MSE (median <2.5%) as well as AIC (median -1.726), as shown in Fig 10. Also, Pathway 3 gave the lowest MSE (median <0.5%) when fitted to the whole 10 min of tHb time series at the targeted-region during HD-tDCS (see S4 Table and S2 Fig). Although perivascular K+ is known for the dynamic regulation of cerebral blood flow as modelled for Pathway 3 within NVU in intracerebral arterioles and microvessels, tDCS activated perivascular nerves in the intracranial blood vessels can also release chemical signals [54] that can alter vascular tone [125]. Guhathakurta and Dutta [27] postulated that tDCS electric field spread in the highly conductive CSF that can directly affect the pial arteries and penetrating arterioles that contain perivascular nerves [28] within their adventitial layer–see Fig 2. In this study, we found that the primary mechanism of transient action for the HD-tDCS-induced CVR between 0.01 and 0.05 Hz is the perivascular pathway within NVU that can also determine the local pial vessel diameter [110]. Here, different tDCS perturbation pathways (shown in Fig 6) have different response time which is crucial for the phenomenological modelling [40] as illustrated by their minimal realization transfer functions (Table 3) in Fig 11A. For example, tDCS perturbation via the synaptic K+ (i.e., Pathway 1) had the slowest effect on vessel circumference. Our nested model evaluation was based on Chi-Square Goodness-of-Fit that determines the quality of fit [101] in terms of Chi-Square difference test [117]. The reduced dimension grey-box linear model (Table 3) for the Pathway 3 with 9 poles and 2 zeros (all free parameters) provided the best Chi-Square Goodness-of-Fit of 0.0078. Our results supporting Pathway 3 is important since recent studies showed that tDCS-induced alterations in cerebral CBF could only be partially related to the cortical excitability changes [9]. Here, our mechanistic investigation of the vascular response to tDCS using a physiologically constrained linear model provided novel evidence supporting the postulated perivascular tDCS effects in intracerebral arterioles and microvessels. Then, postulated elevation in extracellular K+, especially following long term stimulation, can initiate a retrograde, propagating, hyperpolarizing SMC signal that dilates upstream arterioles including pial vessels to increase local blood flow.

Our physiologically constrained linear model for Pathway 1 considered tDCS perturbation of synaptic mediators, which then perturbed the vessel circumference's effect via astrocyte and perivascular compartments through nested neurovascular dynamics (see Fig 6). Pathway 1 is based on conventional neurovascular coupling mechanism under the effect of tDCS current density on the neurons that release K+, where neuronal activity drives the hemodynamic response. Then, Pathway 2 considered the change in the membrane potential of the astrocytic compartment due to tDCS current density, which then drives the vascular response (vessel circumference). Pathway 3 considered modulation of the K+ in the perivascular space by tDCS current density. Pathway 4 considered the direct influence of tDCS current density on the

vasculature's smooth muscle cells leading to the vascular response (vessel circumference). Here, Pathway 4 was based on prior evidence that the voltage-gated potassium channels on the SMC can respond to the electric field [126,127], especially those in the pial vasculature. We found that the tDCS perturbation Pathway 4 gave the lowest AIC (median -0.915) across all subjects at the contralateral nontargeted-region as shown in Fig 10D. This is postulated to be due to the tDCS current spread in the CSF as shown in the S4C Fig. Voltage-gated potassium channels are also present in the skin vascular smooth muscle cells [128]. In this study using low-density fNIRS, we could not dissociate the CVR of intracranial blood vessels from intracerebral blood vessels that may be possible with diffuse optical tomography (DOT) [129].

Investigation of tDCS activated perivascular nerves altering vascular tone [125] is crucial since pial arteries start the pressure-driven blood pathway to the cortex (reviewed in Schmid et al. [14]) so vasoconstriction effects (e.g. Neuropeptide Y) of the direct electric field on the pial arteries can lead to an initial dip in the blood volume (and tHb) that was found in few healthy subjects in Fig 11. Such vasoconstriction effects can also result via extrinsic perivascular innervation [61] from tDCS effects on peripheral nerves [59,60] that needs further investigation with DOT to dissociate CVR of intracranial blood vessels from intracerebral blood vessels. Here, tDCS effects on tHb via vascular neural network under "neurogenic hypothesis" is in contrast to the "metabolic hypothesis" that causes initial dip (0-10sec) in oxy-Hb while there is a rise in deoxy-Hb (tHb does not change). The grey-box linear model captured the initial dip in the tHb (related to the vessel circumference in the physiologically detailed model) since the grey-box linear model was fitted to the experimental data from all 11 subjects (see S2 Table). The parameterized grey-box linear model derived from a detailed physiological model had many states (see S3 Table) where the numerator and denominator polynomial roots are known as model zeros and model poles, respectively. The model poles and zeros are useful in evaluating a system as their values govern the system's stability and performance. For a stable system, all the model poles must have negative real values. The model zeros are related to the response speed for a given system that captured the initial dip in the tHb concentration with positive zero in the right-half-plane (positive real axis for parameters of transfer function). Specifically, this model zero in the right-half-plane slowed the time response and resulted in the undershoot response. A positive zero adds to the phase lag in a system wherein the response initially becomes negative or changes direction to that of the required direction before converging in the desired steady state. However, such states can have relatively small energy contributions to system dynamics; so, a minimal realization transfer function with reduced-order approximations (see the Methods section for details) for the four pathways provided better insights into the linear time-invariant system that was fitted to 8 subjects (see Fig 11 for the rejected subjects who presented an initial dip in tHb).

In this study, our goal was to investigate the role of various neuronal and non-neuronal pathways leading to the fNIRS-tHb based CVR to tDCS, as demonstrated experimentally by our prior works [66,40], through an objective physiologically constrained grey-box model that is summarized by a block diagram in Fig 6. The physiologically detailed nonlinear model was derived from Witthoft and Karniadakis [45]; however, limitations with low-density fNIRS-tHb human data necessitated model linearization and reduction for adequate model fitting. The physiologically detailed model considered the effect of tDCS via NVU that was represented by seventeen states, i.e., seventeen differential equations (see S1 Text). Then, the grey-box linear models after applying the Model Linearizer (Mathworks, Inc. USA) are shown in the S3 Table. Then, the minimal realization transfer functions were obtained through Model Reducer (Mathworks, Inc., USA) that are shown in Table 3. Here, transfer function order denotes the number of model poles where the reduced dimension grey-box model for the Pathway 1 was represented by a twelfth order system that included a first-order linear filter for the tDCS's

vasoactive effects–see Table 3. The reduced dimension grey-box model for the astrocytic-driven (Pathway 2) and the perivascular K+-driven (Pathway 3) pathways were represented by eleventh and ninth order systems, respectively. For Pathway 4, the input path considered the influence of tDCS current density on SMC gated ion channel current, and the reduced dimension grey-box model for this pathway was represented as a seventh order system. The impulse response function of the Pathway 1, capturing conventional hemodynamic response function, peaked around 5 sec (see Fig 12A) that was comparable to known hemodynamic responses [130]. The impulse response function of the four tDCS perturbation pathways provided insights into the temporal dynamics where the vessel response through an astrocytic pathway or perivascular potassium pathway peaked around 2 sec that was found comparable to known capillary responses [14].

In the current study, we investigated the transient initial (0-150sec) tHb response to tDCS so we did not consider nonlinear calcium dynamics during myogenic smooth muscle activity in the frequency range of 0.05–0.2 Hz [131], including the ~0.1 Hz hemodynamic oscillations in the fNIRS time series [50]. Steady-state very low-frequency oscillations between 0.01 and 0.05 Hz can also originate from arterioles under neurogenic innervation [28,29]. Then, ~0.1 Hz hemodynamic oscillations can be related to the synchronization of the intermittent release of calcium within vascular mural cells including SMC [131] where contractile mural cells are known to generate spontaneous calcium transients. These steady-state vessel oscillations need future investigation in conjunction with electroencephalogram (EEG) since our prior works have found a cross-correlation between log (base-10) transformed EEG band-power (0.5–11.25Hz) and fNIRS oxy-Hb signal in that low frequency (≤0.1Hz) range [50]. Future investigation of ~0.1 Hz oscillatory vessel response vis-à-vis neuronal response (EEG) to tDCS will develop a parameterized coupled oscillator (nonlinear) model for limit cycle behavior. The relation of the ~0.1 Hz oscillatory vessel response vis-à-vis neuronal response may be related to the cortical excitability changes to anodal tDCS [132] due to the involvement of potassium and calcium dynamics [35,133–135] in neurovascular communication that needs further investigation using tACS. Here, unlike tDCS, tACS can lead to physiological entrainment at the frequency of stimulation for system identification that can provide physiological insights based on a physiologically detailed model (see the Eqs 48–49 in the S1 Text). Also, longer duration tDCS is postulated to elevate extracellular K+ that can decrease calcium activity mediated by the inward rectifying potassium channel (see Eqs 36, 41 in the S1 Text) in the mural cells where a combination of tDCS and tACS can be used for system identification of the neurovascular communication [135]. Besides inward rectifying potassium channel, voltage-dependent potassium channel, calcium activated potassium channel, ATP-activated potassium channels are also present in the mural cells that can interact with dilatory stress-induced calcium transients in the mural cells. Therefore, tES effects on the contractile mural cells that encircle the precapillary sphincter [136] at the transition between the penetrating arteriole and the first order capillary may be crucial for intracerebral neurocapillary modulation by tDCS [15]. These can be elucidated with multimodal optical imaging of neuronal, astrocytic calcium, and the hemodynamic neurovascular changes in an animal model for system identification.

In summary, our study presented a preliminary linear systems analysis using a physiologically-constrained grey-box model that was found useful to explore various tDCS perturbation pathways in a lumped NVU model related to fNIRS based CVR to HD-tDCS. Such grey-box linear systems analysis using fNIRS-tHb data from individuals with pathological conditions [36] can elucidate dysfunction in various NVU pathways, e.g., due to the pathological dysfunction. Here, our proposed linear systems analysis using the grey-box model is amenable to pole-zeros analysis of the transfer function for various pathways of the neurovascular system in health, aging, and disease that can also be used to classify the dysfunction in neurovascular

communication [135]. This is crucial since neurovascular coupling dynamics are complicated phenomena in humans, and it can be hard to uncouple neuronal and vascular effects of tES without a mechanistic model-based hypothesis testing. The association between neuronal activity and hemodynamic responses can also be examined through other functional neuroimaging techniques such as perfusion-driven Intravoxel Incoherent Motion, fMRI and PET. Here, the integration of tES with functional neuroimaging modalities holds immense promise for throwing light on the underlying neuromodulation processes of current density effects on the neurovascular tissue. Our study provided a rational model-based approach to capture the hemodynamic response to tDCS with low-density fNIRS that is amenable to clinical translation of tES approaches in cerebrovascular diseases due to its ease of use and low cost [137,138]. Also, fNIRS measurement of Cytochrome-C-Oxidase [139] has been shown feasible that is important to investigate tES effects due to the relation of vascular density and the cytochrome oxidase activity [14].

Limitations of this study included the methodical limitations of the low-density fNIRS technique [140]. The fNIRS signal acquired with optodes placed on the scalp can represent different hemodynamic signal sources (cerebral versus extra-cerebral) and other physiological causes (neuronal versus systemic) that can be evoked by tDCS. Due to the lack of short-separation channels to perform short source-detector regression to remove extra-cerebral hemodynamics, we performed a data-driven principal component analysis to identify the extra-cerebral signal components that explained the most amount of covariance across all the 16 spatially symmetrically distributed fNIRS channels. We also adopted the anti-correlation method [141] (see S2 Table) to confirm fNIRS signal quality since the short-separation channels were not available. We found that the tDCS perturbation Pathway 3 presented the least MSE and AIC, as shown in Fig 10. Here, fNIRS-tHb signal without short separation regression may also have a representation from the superficial pial vessels since back-reflection geometry of the fNIRS measurement renders the signal sensitive to pial vasculature [142]. This need further investigation in the future since DOT methods may be able to delineate tDCS effects on the pial vessels from the penetrating arteriole and the first order capillary that may be crucial for investigating the spatiotemporal aspects of the perivascular modulation by tDCS [15] of the intracerebral versus intracranial blood vessels in the humans. Moreover, detailed physiological model of the perivascular nerves [28] (e.g., neuropeptide Y is an important vasoconstrictor [57] of sympathetic innervation, parasympathetic innervation for vasodilation including calcitonin-gene-related peptide) is necessary to delineate mechanisms of the transmural electrical stimulation. Besides initial dip in few subjects (see Fig 11), differential activation of the oxy-Hb and deoxy-Hb over 0–600 sec (see S2 Table) in the other 8 subjects indicated neuronal activation and "metabolic hypothesis" for the intracerebral blood vessels in the targeted-region. Since we only investigated the initial 150sec of fNIRS-tHb response to tDCS so we did not aim to capture the slower neuroplasticity related changes that are expected following longer-term (0.7 to 2.0 mA over 9–20 minute sessions [20]) stimulation. Limiting to the initial 150sec of fNIRS-tHb response allowed the physiologically detailed nonlinear model to be substantially simplified by model linearization and reduction that removed nonlinear system dynamics which may be necessary for capturing the neuroplastic aftereffects of tDCS. Therefore, our parameterized grey-box linear model is applicable for the initial 0-150sec transient response in the 0.01–0.05 Hz frequency band of the fNIRS-tHb. Our investigation considered only the immediate (<150sec) effects of anodal tDCS with K+ as the main vasoactive agent within lumped multi-compartmental model of NVU. Future studies need to investigate the interactions between perivascular and extracellular K+ and dilatory stress-induced calcium transients in the mural cells vis-à-vis steady-state myogenic smooth muscle activity in 0.05–0.2 Hz during longer duration (>150sec) stimulation. Here, neurovascular communication [135] modulated

by tDCS [50] is postulated to be mediated by the K+ channels, including the inward rectifying potassium channels, in the mural cells.

## Supporting information

**S1 Fig. Simulated grey-box linear model output for the four pathways fitted individually to each of the 11 participants' fNIRS-tHb data (600 seconds from HD-tDCS stimulated region).** Averaged experimental fNIRS-tHb response across all subjects is also shown with a dashed line. (A) Pathway: 1: tDCS modulating vessel response through synaptic potassium pathway, (B) Pathway 2: tDCS modulating vessel response through astrocytic pathway, (C) Pathway 3: tDCS modulating vessel response through perivascular potassium pathway, (D) Pathway 4: tDCS modulating vessel response via the smooth muscle cell pathway.
(TIF)

**S2 Fig. Boxplot of the mean square error (MSE) across all the 11 participants for the grey-box model fitting using experimental fNIRS-tHb data from tDCS-stimulated side for 600 seconds for the four model pathways.**
(TIF)

**S3 Fig. Autocorrelation curves for the residuals and cross-correlation curves between input and residuals for the proposed four pathways.** Plots show the residual analysis of the refined models obtained for the proposed pathways using grey-box model estimation data of 11 volunteers (fitted to initial 150 seconds of tDCS, model outputs presented in Fig 4 of the main manuscript). The plots display the autocorrelation curves for the residuals and cross-correlation curves between input and residuals for the proposed pathways. The confidence interval for the curves are shown by dashed lines. (A) Pathway 1: tDCS modulating vessel response through synaptic potassium pathway. (B) Pathway 2: tDCS modulating vessel response through astrocytic pathway. (C) Pathway 3: tDCS modulating vessel response through perivascular potassium pathway. (D) Pathway 4: tDCS modulating vessel response via the smooth muscle cell pathway.
(TIF)

**S4 Fig. Electric Field in brain tissues.** (A) Tissue segmentation for finite element modeling of the electric field using ROAST: An Open-Source, Fully-Automated, Realistic Volumetric-Approach-Based Simulator For TES. (B) Electric field (V/m) in the brain. (C) Electric field (V/m) in the cerebrospinal fluid (CSF)–note that the magnitude difference with the brain in the color scale. (D) Electric field (V/m) in the grey matter.
(TIF)

**S1 Text. Model Equations for Neurovascular Compartmental Dynamics.**
(DOCX)

**S2 Text Transcranial electrical stimulation induced current density in the neurovascular cortical tissue.**
(DOCX)

**S1 Table. Model Parameters.**
(DOCX)

**S2 Table. Oxy and Deoxy-Hemoglobin Changes at the Stimulated Region.** Normalized Oxy-hemoglobin (oxy-Hb), Deoxy-hemoglobin (deoxy-Hb) and Total Hemoglobin (tHb) obtained from NIRS channel in tDCS stimulated region for all the 11 participants. *subjects

with >-0.5 Correlation Coefficient between Oxy-Hb & Dxy-Hb.
(DOCX)

**S3 Table. System Identification.** System identification using experimental data (total hemoglobin changes from stimulated hemisphere) from 11 subjects (Sub). * subjects with >-0.5 Correlation Coefficient between Oxy-Hb & Dxy-Hb.
(DOCX)

**S4 Table. Mean-square error (MSE) for the grey-box model analysis using experimental fNIRS-tHb data from tDCS-stimulated side for 600 seconds of each subject for the four model pathways.** * subjects with >-0.5 Correlation Coefficient between Oxy-Hb & Dxy-Hb.
(DOCX)

**S5 Table. Median, mean, and the standard deviation of the mean-square error (MSE) for the grey-box model analysis using experimental fNIRS-tHb data from tDCS-stimulated side (targeted region) and contralateral side (nontargeted region) for 1–150 seconds of each subject for the four model pathways.** Subjects P3, P4, P10 have >-0.5 Correlation Coefficient between Oxy-Hb & Dxy-Hb.
(DOCX)

**S6 Table. Median, mean, and the standard deviation of the Akaike information criterion (AIC) for the grey-box model analysis using experimental fNIRS-tHb data from tDCS-stimulated side (targeted region) and contralateral side (nontargeted region) for 1–150 seconds of each subject for the four model pathways.** Subjects P3, P4, P10 have >-0.5 Correlation Coefficient between Oxy-Hb & Dxy-Hb.
(DOCX)

## Acknowledgments

The authors would like to acknowledge the help of Ms. Mehak Sood, Mr. Utkarsh Jindal, and Dr. Pierre Besson for the human study.

## Author Contributions

**Conceptualization:** Anirban Dutta.

**Data curation:** Yashika Arora.

**Formal analysis:** Yashika Arora, Anirban Dutta.

**Funding acquisition:** Shubhajit Roy Chowdhury, Stephane Perrey, Anirban Dutta.

**Investigation:** Yashika Arora, Anirban Dutta.

**Methodology:** Yashika Arora, Mitsuhiro Hayashibe, Shubhajit Roy Chowdhury, Anirban Dutta.

**Project administration:** Shubhajit Roy Chowdhury, Anirban Dutta.

**Resources:** Shubhajit Roy Chowdhury, Stephane Perrey, Anirban Dutta.

**Software:** Yashika Arora, Pushpinder Walia.

**Supervision:** Shubhajit Roy Chowdhury, Anirban Dutta.

**Validation:** Yashika Arora, Mitsuhiro Hayashibe, Shubhajit Roy Chowdhury, Anirban Dutta.

**Visualization:** Yashika Arora, Anirban Dutta.

**Writing – original draft:** Yashika Arora, Anirban Dutta.

**Writing – review & editing:** Mitsuhiro Hayashibe, Makii Muthalib, Shubhajit Roy Chowdhury, Stephane Perrey, Anirban Dutta.

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
