## [Decision Letter · Decision Letter 0]

19 May 2021

Dear Prof. Dutta,

Thank you very much for submitting your manuscript "Grey-box modeling and hypothesis testing of functional near-infrared spectroscopy-based cerebrovascular reactivity to anodal high-definition tDCS in healthy humans" for consideration at PLOS Computational Biology.

As with all papers reviewed by the journal, your manuscript was reviewed by members of the editorial board and by several independent reviewers. In light of the reviews (below this email), we would like to invite the resubmission of a significantly-revised version that takes into account the reviewers' comments.

We cannot make any decision about publication until we have seen the revised manuscript and your response to the reviewers' comments. Your revised manuscript is also likely to be sent to reviewers for further evaluation.

Sincerely,

Hugues Berry

Associate Editor

PLOS Computational Biology

Lyle Graham

Deputy Editor

PLOS Computational Biology

Reviewer's Responses to Questions

**Comments to the Authors:**

Reviewer #1: Review is uploaded as a document

Reviewer #2: The author proposes a comprehensive review of NVC computational model and the evidence of modulatory consequences of tDCS on blood vessels. The paper is well written and easy to read, however, there are some points that need to be resolved. Here are some comments which may help the author to further improve the manuscript.

General Comments:

1- The study of NVC mechanisms requires concomitant measurements of changes in both neuronal activity and hemodynamic signals in response to a stimulus. In this respect, pyramidal cells have been identified as “neurogenic hubs” of NVC in the somatosensory cortex [1], and astrocytes have emerged as key players in the communication between activated neurons and blood vessels as recently reviewed [2]. tDCS-induced electric field effects on the hemodynamics via neuronal, astrocyte, and cerebral blood vessels, composed of pericytes, smooth muscle cells and endothelial cells.

[1] Lecrux, C., Toussay, X., Kocharyan, A., Fernandes, P., Neupane, S., Levesque, M., Plaisier, F., Shmuel, A., Cauli, B., Hamel, E., 2011. Pyramidal neurons are “neurogenic hubs” in the neurovascular coupling response to whisker stimulation. J. Neurosci. 31, 9836–9847.

[2] Cauli, B., Hamel, E., 2018. Brain perfusion and astrocytes. Trends Neurosci. 41, 409–413.

The authors postulate “that the immediate hemodynamic response9 based on CVR to short duration tDCS can provide a marker of blood vessels' capacity to dilate that can be hampered in various cerebrovascular diseases36,37. However, longer duration (>9 min) tDCS will induce neuroplastic changes24 where non-linear bidirectional neurovascular interactions38 can make the mechanistic models too complex for fNIRS data fitting and hypothesis testing.”

To which compartment would this short duration tDCS be linked ?

2- The model selection is based on the fitting error which need to be well justified in the manuscript, especially considering the minimization of fitting error will lead to overfitting. How do authors consider overfitting?

Other comments

Abstract:

Context:

1--

We found that the grey-box model pathway from perivascular potassium to the vessel circumference (Pathway 3) presented the best fit to fNIRS tHb time series with the least mean square error (MSE, median < 2.5%). Then, minimal realization transfer function with reduced-order approximations of the grey-box model pathways was fitted to the average tHb time series. Pathway 3, with nine poles and two zeros (all free parameters), provided the best Chi-Square Goodness of Fit of 0.0078.

2--

Therefore, our study provided a sound systems biology approach to investigate the hemodynamic response to tDCS that needs further investigation in health, aging, and disease. Future studies can leverage transcranial alternating current stimulation for frequency-dependent physiological entrainment of relevant pathways, e.g., oscillations driven by nonlinear calcium dynamics, that can provide additional insights based on our grey-box modeling approach.

--

Comment 1: The logic from the two paragraphs (1 and 2) can be applied to any pathway, meaning if we replace Pathway 3 by Pathway N, the contribution of the study claimed by the red sentence will always work. Please describe the contribution of finding on Pathway 3 explicitly. In other words, how do authors justify that it is indeed by finding Pathway 3 which best fit the tHb so that leading the conclusion about their model is a sound systems biology approach to investigate the hemodynamic responses to tDCS?

Introduction

Comment 2: Authors said “Data-driven black-box systems approaches provide a correlate of neural and hemodynamic response at an abstract level under the assumption of neurovascular coupling at the cellular level; however, such black-box systems approaches do not aim to explicitly capture the underlying cellular mechanisms of action. ”

I agree with the claim and love the idea of using an explicit generative model. However, using a more complex model(s) with biological meaningful parameters and relationships is not free. Sophisticated model requires more sophisticated evaluation methods, not just on how well the simulated data fit with real data. As stated by authors, “a deeper understanding of the signaling pathways inside the NVU is essential for a mechanistic understanding”, this means the goal is to infer the understanding of the pathway, not to select which pathway provides the best fit. There is no best model, and each model is assumed and justified based on our current knowledge. Therefore, if the goal is to achieve a mechanistic understanding, we may not want to select the best model, but assume these models are all reasonable and then compare the models’ behaviour on data fitting to understand the research question from the comparison itself instead of selecting the “best model”. We can step back a bit by saying, let us believe pathway3 is the best, then how do authors justify the parameters value and, more importantly, to understand the contribution of each parameter to the final data fitting?

The authors conducted excellent works on reviewing the models that are available from literature and justified the parameters selection and provided the final values in Supplementary. However, if the parameter values can be fixed from the literature review, then it means we have already understood enough this these pathways, so then how do we justify the motivation of this study? In other words, how do authors justify the potential bias on parameter value selection on the final fitting quantification?

Comment 3: The authors talked about the initial dip in the last paragraph of the introduction. However, what is the relationship to this study?

Throughout the manuscript, the duration of interest for fNIRS signal was 0-150s and the time scale of the plot was 0 – 10 minutes, how can we see the dips?

Results

Comment 4: Authors claimed that “In most of the participants, the time course of changes in tHb in the region within the boundary of the HD-tDCS montage showed a steep initial increase compared to the contralateral non-stimulated sensorimotor area that showed a decrease in some participants, which may be related to inter-hemispheric inhibition.”

1) We understand the variability of fNIRS data, but “most of the participants” is not clear. Please state the portion, even if it is based on visual inspection.

2) How do authors justify it is within the boundary? From Figure 6 and the method section, the boundary channels are not even included in the data analysis. (please see the detailed comments in the method section).

3) Most importantly, in Figure 1,

a) each tHb time course started from a different value. Did authors perform baseline correction? If not, how to justify the statement of decrease and increase? Since fNIRS is a relative measurement, please clarify what the reference for these changes is. This is also seen in Supplement D, both HbO and HbR started from different values, and no baseline is shown.

b) since the time course is averaged among subjects, what is the variance? Could the authors please show the error bar?

c) it is nice to compare with the non-stimulated region, however, why not plot them in the same figure for better visual comparison?

4) could the authors indicate in each figure where is the increase and decrease or simply mention the duration of them in the caption? Because the signal just fluctuates and there is an increase/decrease everywhere.

Comment 5: I might be wrong for this point, but the authors stated, “Figure 2(B) shows that physiologically detailed neurovascular coupling pathways generated low-frequency oscillations in the frequency range of 0.05 – 0.2 Hz, driven by nonlinear calcium dynamics in various compartments, during the time course of the vessel response”

Looking at this figure, I counted 3, 16, 5, 7 periodic components for pathways 1 to 4, respectively. Within 10 minutes, the corresponding frequency is 0.005, 0.026, 0.008 and 0.012Hz, only pathway 2 and 4 is within filter band 0.05 – 0.2Hz or the band of interest 0.01-0.05Hz explained later in method. Could the authors please clarify this?

If these fluctuations are not the frequency of interest, then what is the difference between 4 pathways after ignoring these components?

Comment 6: In Figure 1, all time courses in As started from 0 value, but all in Bs started from 0.4. Could the authors please explain why?

In the caption, “The average fNIRS-tHb response across all participants is also shown with a dashed line.” Is this a dashed line representing the real data? Or it is averaged from all color coded curves from simulation?

It was a bit confusing since in the legend, dashed line just followed each subject’s line, it led me to think dashed line is just averaged from all individuals.

Comment 7: MSE is shown in Table1A and Table1B for stimulation and non- stimulation side.

1) please report the error of the mean (i.e sd) even if one can find it from figure 4.

2) The smallest MSE in the stimulation side is Pathway3 which is the core result of this manuscript. Let us first ignore the justification of using MSE to select the model (which may lead to overfitting), the smallest MSE in non-stimulation case is Pathway 1, and the value is similar to Pathway 2 and 4 in simulated case. Then how do we infer the fact that Pathway1 is the best fit in non-stimulation case?

Comment 8: Authors claimed that “The grey-box model simulated output of all the four pathways demonstrated an initial dip in the tHb concentration (related to the vessel circumference in the physiologically detailed model) which was based on the experimental data (see Supplementary materials, section D: Oxy and Deoxy-Hemoglobin Changes at the Stimulated Region).”.

1) I could not find the initial dip in the tHb results in Supplementary D. Could authors please indicate them in the figure? I do see the dips in simulations.

2) please show the error bar (Supplementary D) in the time course plot since it is averaged among channels.

Comment 9: there is no statistics comparison of boxplot in figure 4. I am not insisting that authors have to show significant comparison but at least some level of statistics comparison are necessary.

1) Especially, pathway 1 has a bit high MSE but clearly a much smaller variance than pathway 3. Then can we choose pathway 1 instead of 3?

2) Moreover, in the non-stimulation side, pathway 1 also provided good MSE. If the authors prefer involving the non-stimulation side as a control and it might be fair to select pathway3 since it did not provide low MSE in the non-stimulation side as pathway1. However, how do we justify the use of the contralateral side as a control?

3) We know M1 and S1 both have connections to the contralateral side M1 and S1. In early fNIRS/TMS studies, due to hardware difficulties of stimulating and detecting on the same side. People did stimulate M1 and apply fNIRS to probe the contralateral M1, and used the connectivity to rationalize the investigation [3]. In contrast, in this manuscript, the authors use the contralateral side as a control. Then how do we justify the eligibility of the contralateral side as a control? Maybe using other regions for control is better (see comments for method).

[3] Chiang TC, Vaithianathan T, Leung T, Lavidor M, Walsh V, Delpy DT. Elevated haemoglobin levels in the motor cortex following 1 Hz transcranial magnetic stimulation: A preliminary study. Exp Brain Res 2007;181:555–60. https://doi.org/10.1007/s00221-007-0952-x.

Comment 10: Authors listed the parameters for different models in both main context and supplementary. As one can see different pathway has a different number of parameters. How do authors justify the influence of the number of parameters on model fitting results? As we know, this factor influences a lot the fitting performance (more parameters tend to overfit).

Discussion

Comment 11: Authors mentioned: “We found that Pathway 3 presented the least MSE, as shown in Figure 4, so the fNIRS- tHb signal is postulated to have a significant representation from the superficial pial vessels.”

Regardless of the justification of statistics significant test on evaluating a hypothesis, please avoid saying “significant” without doing a test in Figure 4.

Comment 12: The authors stated that: “In this study, our goal was to investigate the role of various neuronal and non-neuronal pathways leading to the CVR to tDCS, as demonstrated experimentally by our prior works42,60,61, through an objective physiologically constrained grey-box model”.

First of all, the authors indeed conduct detailed model selection and explanation based on literature. The models involved in the manuscript also covered the concept of neuronal and non-neuronal pathways. However, the tHb data from fNIRS, is not purely neuronal related signal. Although the authors used PCA to remove the physiological part of the fNIRS signal, but it is not guaranteed to clean all of them. We admit there is no way in continuous-wave fNIRS to remove the physiological noise completely, but such confounding plays a big role in justifying the results and rationale of fitting to fNIRS data for model checking, especially the criteria is MSE. Please also see the details in the comments of the method section.

Method

Comment 13: Could authors please check figure 6 carefully? There are several confusions:

1) for each hemisphere, A has 6 detectors and 2 sources, but in results and data analysis and figure caption, only 2 detectors were used. Could the authors please explain why the other 4 detectors (F2, F6, P2 and P6 on right hemi) were not used? We understand that they may not give a good response since they might be far from to stimulation site, but then please do not mention them directly in the figure.

2) how come the sensitivity value was negative? The color bar has the highest value of 0

Comment 14: Authors mentioned, “Due to the lack of short-separation channels to perform short source-detector regression to remove extra-cerebral hemodynamics, we performed data-driven principal component analysis (PCA) to identify the signal components that explained the greatest amount of covariance across all the spatially symmetrically distributed 16 channels”

How 16 channels are counted? The previous paragraph clearly said 4 channels on each side. If all channels in Figure 6 (A) were used, then they are indeed 16 channels, but the rest channels are not involved in the results inferences. Could the authors please clarify this?

It is always good to find a way to remove extra-cerebral hemodynamics even if no short-separation channels are available from the hardware side. Considering the importance of the potential confounding from extra-cerebral hemodynamics on the justification of the model evaluation of this manuscript, it would be nice if the authors could compare the results with/without PCA, to show at least there is a difference on the model evaluation side. For instance, without PCA procedure, the results on model evaluation should be different, or the difference between pathways should be diluted since extra-cerebral hemodynamics are involved.

Alternatively, the authors could have performed analysis on the component from PCA that was believed containing extra-cerebral hemodynamics, and then compare it with the current results. It has been shown in [4], without careful analysis, TMS induced hemodynamics measured on the scalp could have been very similar to the one when stimulating the shoulder.

A recent review [5] of TMS/fNIRS also addressed the necessity of using short-channel to remove the extra-cerebral hemodynamics for more reliable results. We understand tDCS may not induce fNIRS signal changes in the scalp (by contracting muscles) as high as TMS. Pleases note that this does not mean we believe authors have to have short-channels and does not mean we believe that short-channel is the best approach, but it means no matter what approach is used to remove the confounding, it is necessary to prove it worked on the final results, meaning with/without it would lead to different results that are as expected.

[4] Näsi T, Mäki H, Kotilahti K, Nissilä I, Haapalahti P, Ilmoniemi RJ. Magnetic-stimulation-related physiological artifacts in hemodynamic near-infrared spectroscopy signals. PLoS One 2011;6. https://doi.org/10.1371/journal.pone.0024002.

[5] Curtin A, Tong S, Sun J, Wang J, Onaral B, Ayaz H. A systematic review of integrated functional near-infrared spectroscopy (fNIRS) and transcranial magnetic stimulation (TMS) studies. Front Neurosci 2019;13. https://doi.org/10.3389/fnins.2019.00084.

Comment 15: The authors said “The neurovascular and neurometabolic coupling-related hemodynamic response should lead to an initial increase in deoxy-Hb and an equal decrease in oxy-Hb, as shown by Devor et al.10. Then, the blood volume should start to increase about 600ms following the neural stimulus10. Such differential activation of oxy-Hb and deoxy-Hb was found in the vicinity of 4x1 HD-tDCS electrodes: C5-CP3, C1-CP3, C5-FC3, and C1-FC3, that indicated neurovascular coupling related hemodynamic response by HD-tDCS current density at the ipsilateral primary motor cortex.”

We do not understand why always mention the initial dip, and we did not see the dip in fNIRS results. How can we see 600ms changes in a figure that has a time scale of 10 minutes? Could the authors please show a zoom-in figure to prove the finding of an initial dip in fNIRS?

Comment 16: In general, the font in the figures is a bit small. Please make them larger if not too much work. Otherwise, they are readable after zooming.

Reviewer #3: My questions and comments on your manuscript are described in the attached file. Sincerely.

**Have the authors made all data and (if applicable) computational code underlying the findings in their manuscript fully available?**

Reviewer #1: **No: **The authors did not include any link to the code repository

Reviewer #2: Yes

Reviewer #3: **No: **The data illustrated in Fig.1 (tHb responses of participants) but I maybe mistaken not seeing them in the editorial system

PLOS authors have the option to publish the peer review history of their article (what does this mean?). If published, this will include your full peer review and any attached files.

Reviewer #1: No

Reviewer #2: No

Reviewer #3: No
---

## [Decision Letter · Decision Letter 1]

28 Aug 2021

Dear Prof. Dutta,

We are pleased to inform you that your manuscript 'Grey-box modeling and hypothesis testing of functional near-infrared spectroscopy-based cerebrovascular reactivity to anodal high-definition tDCS in healthy humans' has been provisionally accepted for publication in PLOS Computational Biology.

Best regards,

Hugues Berry

Associate Editor

PLOS Computational Biology

Lyle Graham

Deputy Editor

PLOS Computational Biology

Reviewer's Responses to Questions

**Comments to the Authors:**

Reviewer #1: The authors have sufficiently addressed the comments and added relevant discussion. I recommend this paper to be accepted.

Reviewer #2: 1) The authors clarified that they selected the best model by the trade-off between MSE and AIC saying in the introduction: "Here, grey-box linear systems identification was performed following model linearization of a physiologically detailed NVU model to find appropriate model fit (mean square error) and model complexity (Akaike information criterion[17]) to the tDCS-evoked change in the fNIRStHb data.”

I would prefer to use "out-sample prediction error" or simply "prediction error" rather than "model complexity". I agree that AIC is related to model complexity but this is just an intuitive interpretation of the AIC equation. The criteria itself is meant to measure the prediction error of the model when using new samples, not to measure the model complexity.

A side note may be too aggressive: AIC is indeed a great achievement from Akaike who find a simple and elegant analytical approximation to calculate out-sample prediction error, but it is now more or less a historical method (I often see people especially statisticians being mean to AIC on Twitter or blogs) considering the derivation of it requires few assumptions (e.g., the posterior distribution must be a multivariate Gaussian) that are often not being hold in practice. There are other information criteria metrics (e.g., WAIC) and numerical solutions such as cross-validation that should provide more accurate prediction error estimation.

2) Comments 3, 8 and 11: I may have limited understanding of the term "initial dip", which for me is often referred to as the decrease of hemodynamic responses during the very first few seconds (people even talk about it within the first 1s) after the onset. The authors revised with details and pointed out the initial dip in Fig.11, but for them, the dip can last to 150s after the task onset. I am not sure 150s after the onset is too long to be defined as an "initial dip". It is minor since it's just terminology describing their observations on the fluctuations of hemodynamic responses.

**Have the authors made all data and (if applicable) computational code underlying the findings in their manuscript fully available?**

Reviewer #1: Yes

Reviewer #2: None

PLOS authors have the option to publish the peer review history of their article (what does this mean?). If published, this will include your full peer review and any attached files.

Reviewer #1: No

Reviewer #2: No

---

## [Editor Report · Acceptance letter]

10 Sep 2021

PCOMPBIOL-D-21-00317R1 

Grey-box modeling and hypothesis testing of functional near-infrared spectroscopy-based cerebrovascular reactivity to anodal high-definition tDCS in healthy humans

Dear Dr Dutta,

I am pleased to inform you that your manuscript has been formally accepted for publication in PLOS Computational Biology. Your manuscript is now with our production department and you will be notified of the publication date in due course.

With kind regards,

Olena Szabo
